# Honour, competition and cooperation across 13 societies

Shuxian Jin [1] ✉, Angelo Romano [2,20], Vivian L. Vignoles [1,20], Alexander Kirchner-Häusler[1,3], Rosa Rodríguez-Bailón [4], Susan E. Cross [5], Meral Gezici Yalçın [6], Charles Harb[7], Shenel Husnu[8], Keiko Ishii [9], Panagiota Karamaouna[10], Konstantinos Kafetsios[11,12], Evangelia Kateri [10], Juan Matamoros-Lima[4,13], Rania Miniesy[14], Jinkyung Na [15], Stefano Pagliaro[16], Charis Psaltis[17], Dina Rabie [18], Manuel Teresi[19], Yukiko Uchida[3] & Ayse K. Uskul [1] ✉

Effectively addressing societal challenges often requires unrelated individuals to reduce conflict and successfully coordinate actions. The cultural logic of 'honour' is frequently studied in relation to conflict, but its role in competition and cooperation remains underexplored. The current study investigates how perceived normative and personally endorsed honour values predict competition and cooperation behaviours. In an online experiment testing preregistered hypotheses, 3,371 participants from 13 societies made incentivized competition decisions in a contest game and cooperation decisions for coordination in a step-level public goods game. Perceived normative honour values were associated with greater competition and greater cooperation at both societal and individual levels. Personally endorsing values tied to defence of family reputation was associated with greater coordinative efforts, whereas endorsing self-promotion and retaliation was associated with weaker engagement in coordination. These findings highlight the role of honour as a cultural logic (in its different forms) in shaping competition and cooperation across societies.

Social interactions frequently involve conflicts of interest between individuals, where the actions available to individuals (for example, competition or cooperation) and the outcomes they might receive (for example, zero-sum or positive-sum) can vary extensively[1–3]. For instance, in formally structured contests where individuals compete for status or limited resources, the outcomes can be zero-sum—meaning that a gain for one party directly translates into a loss for another[4]. In contrast, situations where individuals coordinate to achieve a common good at a personal cost often involve positive-sum outcomes, where the collective gain for all parties exceeds what any one of them could achieve independently[5]. Understanding these different types of interactions is essential for addressing societal challenges, such as mitigating conflict and fostering efficient coordination among unrelated members of society.

Past literature has taken different perspectives on studying competition and cooperation. Some researchers categorize these behaviours as representing two extremes of a singular behavioural spectrum[4,6], while others consider them as entwined components harmoniously coexisting or even being positively related in conflicting-interest situations[7–9]. Empirical research has increasingly investigated when and why individuals compete and/or cooperate with others, though largely in separate studies, both within and across cultural contexts[10–15]. Recent cross-cultural research, containing evidence from non-Western regions, investigated a range of ecological, social and institutional factors that may account for cross-cultural variation in competition and/or cooperation[13,16,17]. Honour, a relevant yet underexplored cultural concept, is particularly prevalent in certain non-Western regions (for example, the Middle Eastern and North African societies)[18–21] and may

---

**Table 1 | Summary of descriptives**

| Society | N | Language | Percentage female (%) | Mean age (s.d.) | % Comp (E) | % Coop (E) | PNH (O) | PNH (F) |
|---|---|---|---|---|---|---|---|---|
| Egypt | 270 | Arabic | 50.38 | 40.78 (14.00) | 69.45 (60.20) | 66.54 (62.20) | 6.03 | 0.41 |
| Greece | 255 | Greek | 49.61 | 40.59 (13.76) | 64.86 (57.25) | 64.15 (60.41) | 5.29 | 0.19 |
| Greek Cypriot community | 269 | Greek | 50.93 | 41.22 (14.20) | 65.72 (59.55) | 64.13 (62.88) | 5.35 | 0.48 |
| Italy | 270 | Italian | 50.37 | 41.14 (14.21) | 62.34 (57.42) | 62.57 (60.75) | 5.04 | −0.09 |
| Japan | 261 | Japanese | 49.23 | 41.56 (14.91) | 64.12 (57.09) | 57.06 (56.44) | 4.50 | −0.34 |
| Lebanon | 250 | Arabic | 53.01 | 39.25 (12.83) | 61.17 (50.36) | 59.69 (56.84) | 5.64 | −0.08 |
| Morocco | 260 | Arabic | 49.22 | 39.81 (13.15) | 67.66 (59.25) | 63.71 (59.56) | 5.66 | 0.55 |
| South Korea | 271 | Korean | 49.82 | 41.21 (14.61) | 62.00 (55.50) | 60.13 (60.06) | 4.89 | 0.05 |
| Spain | 249 | Spanish | 48.19 | 40.81 (14.30) | 62.76 (54.73) | 61.55 (58.20) | 4.98 | −0.16 |
| Turkish Cypriot community | 245 | Turkish | 49.80 | 40.32 (14.46) | 59.42 (57.61) | 59.62 (59.62) | 5.05 | 0.17 |
| Türkiye | 260 | Turkish | 50.77 | 40.72 (14.01) | 67.62 (61.79) | 66.66 (64.45) | 5.50 | 0.15 |
| United Kingdom | 255 | English | 49.80 | 41.47 (15.79) | 62.51 (55.69) | 60.95 (56.14) | 4.45 | −0.60 |
| USA | 256 | English | 51.01 | 41.33 (16.25) | 62.22 (55.68) | 61.42 (57.77) | 4.44 | −0.72 |
| **Total** | **3,371** | | **50.16** | **40.79 (14.36)** | **64.03 (57.13)** | **62.20 (59.68)** | **5.14** | |

% Comp (E), percentage of competitive investments (percentage of expectations of others' competitive investments); % Coop (E), percentage of expectations of others' cooperative investments); PNH (O), societal mean of perceived normative honour values (observed score); PNH (F), societal-level perceived normative honour values (factor score). See Supplementary Table 35 for more summary information on the age range, parents' education level, subjective social status, ethnicity and living environment (for example, urban or rural) of the sample from each society.

act as an important cultural logic shaping how individuals navigate conflicts of interest between the self and others.

Honour can be understood as the value of a person in their own eyes and in the eyes of others[22]. To be honourable, individuals must actively express certain traits or behaviours to claim honour and gain recognition and respect from others in their social environment[23–25]. Recently, honour has been studied as a cultural logic comprising shared beliefs, values, norms and practices that cohere around the central theme of pursuing honour[26]. This cultural logic tends to emerge in harsh, competitive environments characterized by status inequality and instability as well as historically weak institutions[27–29]. In these environments, individuals are likely to develop strategies to protect their safety and resources, as well as those of their close ingroups such as family members, through personal actions. A reputation for toughness and strength is adaptive because it can deter competitors and prevent being exploited in the future[26,28,30]. Individuals' willingness to retaliate or even pre-emptively defend themselves, securing a tough reputation, can be selected as an important survival strategy and thus become normative in these environments[31]. Moreover, individuals may engage in similar actions to defend the honour of their close others or affiliated social groups (typically family members)[32]. However, the pursuit of honour seems to risk escalating unnecessary conflict, especially among unrelated individuals. Past literature has documented that honour-related norms and behaviours can foster conflict responses such as violence, aggression and honour-related crimes[28,33–36].

To study how the cultural logic of honour may shape both competition and cooperation, we employed two separate incentivized economic games that may provide different opportunities for the expression of honour-related values and norms[37,38]. Economic games are highly structured situations with formal rules and unambiguous outcomes, which are nonetheless widely used to study human judgement, decision-making and behavioural choices that may transfer into everyday life[37,39]. We examined how individuals' behaviour in these games may be predicted by honour values on multiple levels: societal-level variation in honour culture (that is, the effects of living in societies where honour values are more or less prevalent)[40],

individual-level variation in perceived societal honour norms (that is, the effects of perceiving honour values as more or less normative in one's society—also known as intersubjective culture)[21,41,42] and individual-level variation in personal honour values (that is, the effects of personally internalizing cultural values of honour more or less)[26].

Contest games are formally structured conflict situations in which one can be better off only at the cost of the other, and one risks being exploited if losing to one's opponent[43,44]. These games have been used to study informal and formal types of competition, as they model conflict situations that result in zero-sum outcomes (for example, public debates, sports competitions and leadership elections). In societies more strongly characterized by a cultural logic of honour, competition can serve as an important means for achieving or maintaining honour, while failure to compete may be perceived as a sign of weakness, leading to potential losses of reputation and social status for individuals (and their close associates, such as family members)[45,46]. We thus expected that members of societies where honour values are more prevalent would exhibit higher levels of competition (H1a) and expectations about interpersonal competition (H1b). At the individual level, we hypothesized that the more individuals perceive honour values as being societally prevalent, the more likely they are to engage in competitive actions themselves (H2a) and expect unrelated others to adopt similar strategies, expressing toughness and competing to promote oneself or prevent losing resources (H2b). Moreover, individuals who more strongly endorse honour values may be more likely to adopt strategies expressing strength and toughness in front of others by engaging in more competitive actions (H3)[47].

Step-level public goods games (PGGs) model situations where individuals can cooperate to achieve better collective outcomes at the risk of wasting personal efforts if coordination fails (for example, building a neighbourhood security system or communal infrastructure)[3,5]. Compared with continuous PGGs, the step-level form transforms the cooperation game into a social coordination problem that aligns self-interests more closely with collective interests and increases the likelihood of cooperation[15]. Investing in coordinating the successful provision of a public good does not necessarily signify weakness.

Unlike contest games where one can benefit only by imposing a cost on others, step-level PGGs give individuals the choice between extending benefits to others at a personal cost and refraining from doing so[48]. The latter type of game enables individuals to express their benevolence, generosity, hospitality and politeness, which may enhance their own honour and that of their close ingroup[26,30,49,50]. However, the inherent risk of wasting coordinative efforts may place individuals in a 'sucker's situation' if others do not cooperate, potentially suggesting a negative link between honour and cooperation[51,52]. We therefore did not formulate specific hypotheses but explored the relationship between honour and cooperation.

The experiment reported here involved a sample of 3,371 participants stratified by age and gender from 13 societies (see Table 1 and Supplementary Table 35 for more demographic information) to test our preregistered hypotheses (https://osf.io/r9atc) and examine further research questions about how perceived normative and personally endorsed honour values relate to competition and cooperation. Participants were recruited online through panel agencies and local research companies (Methods). Nine of the 13 societies—Spain, Italy, Greece, Turkey, Cyprus (both Greek Cypriot and Turkish Cypriot communities), Lebanon, Egypt and Morocco—were in the Mediterranean region, where recent findings have shown that honour values are deeply ingrained in individuals' social worlds, albeit in different forms and to a greater extent in societies further east and/or south within this region[21]. The participants made 12 independent rounds of decisions in two economic games (six rounds per game). Each round was played with a different participant from the participants' own society, whose decision was asynchronously paired after the experiment for payment calculation. We studied interactions among unrelated individuals from the same society to avoid confounding our outcomes with competitiveness between societal ingroup (citizens) and outgroup members (foreigners)[17].

Competition was measured in a contest game where participants could invest their money in an attempt to take away their opponent's money (Fig. 1)[43,44]. If a participant invested more than their opponent, they could take all the money that the opponent did not invest; if both participants invested the same amount (that is, a tie), they would each keep whatever money they had not invested. Cooperation was measured in a coordination game: a step-level PGG with two provision levels (16 and 12 monetary units (MUs)) where participants could attempt to reach the provision levels of the public good by contributing money that would be combined with their partner's contributions (Fig. 1)[53]. A compelling decision rule, potentially rooted in concepts of equity and fairness, is to equally share the cost to meet a provision point (for example, contributing 8 or 6 MUs). Such decisions are often referred to as focal points in coordination games, and the frequency with which individuals make these decisions can reflect their coordinative efforts[5]. After each decision in both games, we asked the participants to indicate their beliefs about their partner's decision, which we used to test H1b and H2b as well as to define further outcomes for exploratory analyses (Methods).

Here we assessed both individual and family (close ingroup) facets of honour because these two facets may have different implications for social interactions within the cultural logic of honour. Specifically, our measure of individual honour focused on valuing certain traits and actions (for example, self-promotion and retaliation (SPR)) to claim honour, whereas our measure of family honour mainly focused on protecting and defending the family's reputation[21,54]. Compared with the family facet, individual honour may be theoretically more relevant for shaping decisions in the dyadic interactions captured in the current study. However, empirical research into the implications of family honour remains limited so far. We sought to contribute to this literature by testing whether the degree to which individuals value defending the honour shared by their family shapes their interactions with unrelated others in their society.

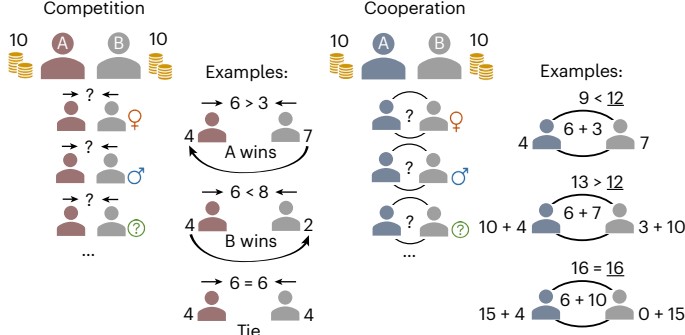

**Fig. 1 | Summary of the design.** In the contest game, participants (red avatar) invested money to attempt to take away the money from their game partner (competition decisions). All invested money would be lost. If a participant invested more than their partner, they could take all the money that their game partner did not invest. However, if both participants invested the same amount, they would each keep whatever money they had not invested. In the step-level PGG, participants (blue avatar) invested money (together with their game partner's investment) to attempt to reach the provision points of the public good (cooperation decisions). The total amount invested by both participants was summed and compared to two provision points. If the total investment reached the first provision point of 12 MUs, each participant received 10 MUs plus any money they had not invested. If the total investment reached the second provision point of 16 MUs, each participant received 15 MUs plus any money they had not invested. In each round, participants faced a different game partner from the same society, with manipulated gender information (male, female or not provided). After data collection, participants' decisions were asynchronously matched with another participant's decisions, on the basis of the manipulated gender information, to compute game payments without deception (see also Methods).

We operationalized the cultural logic of honour through the individual-level measures of personal endorsement of the abovementioned two facets of honour values (referred to as personal values) as well as intersubjective perceptions of how prevalent the two facets of honour values are within each society (referred to as perceived normative values)[41,42]. The society mean of perceived normative honour values across both facets was used to construct a societal-level indicator, characterizing the extent to which a society can be considered a culture of honour (referred to as societal-level honour), ranging in our current samples from 4.44 (USA) to 6.03 (Egypt) (see Table 1 for the scores of all samples). As preregistered, we measured additional variables at the individual level, including beliefs in a zero-sum game[55] and relational mobility[56], and obtained society means to construct societal-level indicators for these variables. These variables may offer additional explanations for competition and cooperation, respectively, and have been shown to vary cross-culturally (see Methods and Supplementary Information sections 3.2.5 and 3.3.5 for more details).

The results revealed that perceived normative honour values were positively associated with competition, cooperation and expectations of these behaviours from others, at both societal and individual levels. Further analyses revealed that perceived normative honour values, particularly defence of family reputation (DFR), were positively associated with coordinative decisions, anticipation of successful coordination and willingness to engage in conditional cooperation. Regarding personal honour values, DFR values were linked to increased cooperative and coordinative efforts, whereas SPR values were associated with reduced efforts in these behaviours.

## Results
### Competition and cooperation
We observed significant differences across societies in competition and cooperation, with between-society variance significantly different from

**a**

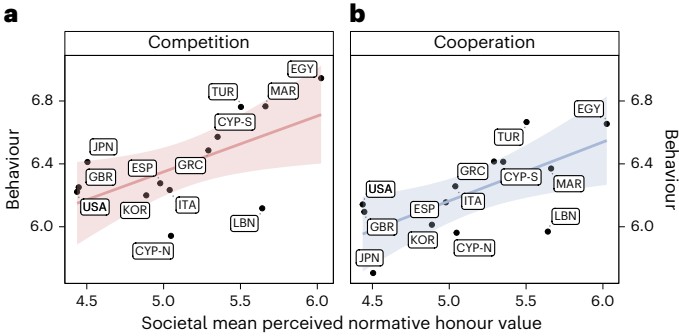

**b**

**Fig. 2 | The relation between societal-level honour (that is, societal mean perceived normative honour values), competition and cooperation.**
**a**, Honour and competition. **b**, Honour and cooperation. Each graph was obtained by regressing the competition or cooperation behaviour on the societal mean perceived normative honour values. The dots represent society-level means and are labelled by country ISO code 3 (Supplementary Table 35). CYP-N indicates the Turkish Cypriot community, and CYP-S indicates the Greek Cypriot community. The shaded area indicates the 95% CI. Societal mean perceived normative honour value (referred to as societal-level honour) was significantly and positively associated with competition (H1a: $\beta = 0.07$, $P = 0.027$) and, surprisingly, also cooperation behaviour ($\beta = 0.08$, $P = 0.013$).

zero for competition ($\chi^2_1 = 31.30$, $P < 0.001$) and cooperation ($\chi^2_1 = 39.80$, $P < 0.001$) (Supplementary Table 3). Consistent with previous findings that competition and cooperation are not bipolar opposites[7,8], we found that competition and cooperation were positively associated both at the societal level (standardized regression coefficient: $\beta_{\text{predicting competition}} = 0.11$; $t_{11} = 3.95$; $P = 0.002$; 95% confidence interval (CI), (0.05, 0.17); $\beta_{\text{predicting cooperation}} = 0.12$; $t_{11} = 3.97$; $P = 0.002$; 95% CI, (0.05, 0.18)) and at the individual level ($\beta_{\text{predicting competition}} = 0.58$; $t_{3354} = 41.51$; $P < 0.001$; 95% CI, (0.55, 0.61); $\beta_{\text{predicting cooperation}} = 0.57$; $t_{3354} = 41.51$; $P < 0.001$; 95% CI, (0.55, 0.60); Supplementary Table 4 and Supplementary Fig. 1).

### Honour and competition

Across 13 societies, societal-level honour was associated with greater competition (H1a: $\beta = 0.07$; $t_{11} = 2.56$; $P = 0.027$; 95% CI, (0.01, 0.13); Fig. 2a and Supplementary Table 5), but not necessarily higher expectations about others' competition (H1b: $\beta = 0.04$; $t_{11} = 1.10$; $P = 0.294$; 95% CI, (−0.04, 0.11); Supplementary Table 6). At the individual level, perceived normative honour values of SPR and DFR were related to higher levels of competition (mixed-effects regression controlling for societal-level honour, partner gender, participant gender, age and game order; H2a: $\beta = 0.05$; $t_{3351} = 2.59$; $P = 0.010$; 95% CI, (0.01, 0.08) (SPR); $\beta = 0.07$; $t_{3351} = 3.45$; $P = 0.001$; 95% CI, (0.03, 0.11) (DFR); Supplementary Table 5) and increased expectations of others' competition (H2b: $\beta = 0.04$; $t_{3351} = 2.11$; $P = 0.035$; 95% CI, (0.003, 0.07) (SPR); $\beta = 0.07$; $t_{3351} = 3.39$; $P = 0.001$; 95% CI, (0.03, 0.10) (DFR); Supplementary Table 6). Individual-level measures of personal honour values across both facets were not associated with engagement in competitive behaviour (H3: $\beta = -0.03$; $t_{3351} = -1.45$; $P = 0.146$; 95% CI, (−0.06, 0.01) (SPR); $\beta = 0.02$; $t_{3351} = 1.15$; $P = 0.251$; 95% CI, (−0.02, 0.06) (DFR); Supplementary Table 5). Robustness checks using factor scores of honour values confirmed these results, with the addition that the positive association between perceived normative honour values of SPR and expectations of others' competition became non-significant (Supplementary Tables 7 and 8).

Next, we explored the potential interaction between individual-level personal honour values and societal-level honour, as the implications of personally endorsing honour values could differ according to the broader cultural logic in one's society. Indeed, we observed a complex pattern for personal values related to DFR ($\beta = -0.03$; $t_{3349} = -2.08$; $P = 0.038$; 95% CI, (−0.07, −0.002)), but no

significant interaction for SPR ($\beta = 0.01$; $t_{3349} = 0.83$; $P = 0.409$; 95% CI, (−0.02, 0.04); Supplementary Table 9). Specifically, the relationship between personal values of DFR and competition was positive in societies with lower societal-level honour but became non-significant as society-level honour increased (see Supplementary Fig. 2 for simple slope analyses). We also explored whether individuals with the same levels of perceived normative and personally endorsed honour values, but inhabiting societies with differing societal-level honour, would differ in their engagement in competition and expectations of others' competition, but we found no support for these contextual effects (competition: $\beta = 0.02$; $t_{13} = 0.64$; $P = 0.533$; 95% CI, (−0.04, 0.08); expectation: $\beta = -0.01$; $t_{12} = -0.20$; $P = 0.843$; 95% CI, (−0.08, 0.07); Supplementary Table 10).

Following the preregistered analysis plan, we tested beliefs in a zero-sum game as a potential additional explanation for competition. Societal mean beliefs in a zero-sum game explained no significant variation in competition beyond societal-level honour ($\beta = -0.03$; $t_8 = -0.87$; $P = 0.411$; 95% CI, (−0.12, 0.06)), and individual-level beliefs in a zero-sum game explained no significant variation beyond personal and perceived normative honour values ($\beta = -0.001$; $t_{2841} = -0.07$; $P = 0.946$; 95% CI, (−0.03, 0.03); Supplementary Table 11). These results were replicated using factor scores of honour values and beliefs in a zero-sum game (Supplementary Table 12). Further exploration of other societal-level indicators theoretically relevant to the cultural logic of honour in relation to competition can be found in Supplementary Information section 3.2.6 (Supplementary Tables 13 and 14).

### Honour and cooperation

Societies characterized by higher mean perceived normative honour values showed higher levels of cooperation ($\beta = 0.08$; $t_{11} = 2.97$; $P = 0.013$; 95% CI, (0.02, 0.14); Fig. 2b and Supplementary Table 15) and expectations of interpersonal cooperation ($\beta = 0.07$; $t_{11} = 2.49$; $P = 0.030$; 95% CI, (0.01, 0.13); Supplementary Table 16). At the individual level, perceived normative values of SPR predicted more cooperation ($\beta = 0.05$; $t_{3351} = 2.78$; $P = 0.005$; 95% CI, (0.01, 0.08); Supplementary Table 15), although they were not associated with expectations of others' cooperation ($\beta = 0.03$; $t_{3351} = 1.91$; $P = 0.056$; 95% CI, (−0.001, 0.07); Supplementary Table 16). Perceived normative values of DFR predicted greater expectation of others' cooperation ($\beta = 0.07$; $t_{3351} = 3.76$; $P < 0.001$; 95% CI, (0.03, 0.11); Supplementary Table 16) but were not associated with participants' own cooperation ($\beta = 0.03$; $t_{3351} = 1.62$; $P = 0.105$; 95% CI, (−0.01, 0.07); Supplementary Table 15). The two facets of personal honour values showed more complex patterns depending on society-level honour values. Overall, personal values of DFR positively predicted cooperation ($\beta = 0.06$; $t_{3351} = 3.00$; $P = 0.003$; 95% CI, (0.02, 0.09); Supplementary Table 15); this positive association was stronger in societies with lower societal-level honour, becoming non-significant as societal-level honour increased ($\beta = -0.04$; $t_{3349} = -2.54$; $P = 0.011$; 95% CI, (−0.07, −0.01); see Supplementary Table 19 and Supplementary Fig. 3 for simple slope analyses). Personal values of SPR did not predict cooperation overall ($\beta = -0.02$; $t_{3351} = -0.95$; $P = 0.342$; 95% CI, (−0.05, 0.02); Supplementary Table 15), but their relationship was negative in societies with lower societal-level honour, becoming weaker or even positive as societal-level honour increased ($\beta = 0.04$; $t_{3349} = 2.67$; $P = 0.008$; 95% CI, (0.01, 0.07); Supplementary Table 19 and Supplementary Fig. 3). The results were similar when we used factor scores for honour values (Supplementary Tables 17–19).

We then explored whether individuals with the same levels of perceived normative and personally endorsed honour values, but inhabiting societies with differing societal-level honour, would differ in their engagement in cooperation and expectations of others' cooperation, but we found no support for these contextual effects (cooperation: $\beta = 0.03$; $t_{12} = 1.02$; $P = 0.327$; 95% CI, (−0.03, 0.09); expectation: $\beta = 0.02$; $t_{13} = 0.69$; $P = 0.506$; 95% CI, (−0.04, 0.07); Supplementary Table 20). As

preregistered, we tested relational mobility as a potential additional explanation for cooperation. Societal mean relational mobility did account for additional variation in cooperation beyond societal-level honour ($\beta = 0.06$; $t_{10} = 2.64$; $P = 0.025$; 95% CI, (0.01, 0.10); Supplementary Table 21), and individual-level relational mobility positively predicted cooperation beyond personal and perceived normative honour values ($\beta = 0.03$; $t_{3350} = 2.38$; $P = 0.017$; 95% CI, (0.01, 0.06); Supplementary Table 21). Yet, these results were not replicated using factor scores of honour values and relational mobility (Supplementary Table 22). Importantly, interpretations of societal-level patterns from the model containing both societal-level honour and societal-level relational mobility as predictors should be made cautiously, given the relatively small number of societies ($N_{society} = 13$), which may have limited the statistical power and generalizability of these findings[57]. Further exploration of other societal-level indicators in relation to cooperation can be found in Supplementary Information section 3.3.6 (Supplementary Table 23).

As preregistered, we conducted secondary analyses of existing meta-analytic and empirical datasets that measured cooperation using prisoner's dilemmas (PD) and continuous PGGs. In these situations, non-cooperation can always yield the best outcome for an individual regardless of what others do. We used societal mean perceived normative honour values retrieved from Study 2 of Vignoles et al.[21] to predict study-level mean cooperation[13] in a meta-regression and individual-level cooperation[16] in mixed-effects models, using data retrieved from previous studies (see Supplementary Information section 3.3.7 for more information). The results showed that societal-level honour did not predict either study-level cooperation rates ($B = 0.06$, $t_{1151} = 0.70$, $P = 0.487$, $\Delta$ pseudo $R^2 = 0\%$; Supplementary Table 24) or individual-level cooperation ($\beta = 0.02$; $t_7 = 0.39$; $P = 0.707$; 95% CI, (−0.11, 0.15); Supplementary Table 26).

The step-level PGG allowed us to analyse individuals' willingness to coordinate by examining the focal point decisions (that is, contributing 8 or 6 MUs). We thus explored the likelihood with which individuals made coordinative decisions to contribute exactly 8 or 6 MUs. Societal-level honour was positively associated with coordinative efforts targeting achieving efficient coordination (that is, contributing 8 MUs) (generalized linear mixed model: odds ratio (OR), 1.14; $P = 0.001$; 95% CI, (1.06, 1.23)), as were individual-level perceived normative honour values of DFR (OR = 1.30; $P < 0.001$; 95% CI, (1.17, 1.45); Supplementary Table 27). Conversely, personally endorsing SPR was negatively associated with the likelihood of contributing 8 MUs (OR = 0.84; $P < 0.001$; 95% CI, (0.77, 0.92); Supplementary Table 27). We found no significant association between societal-level (OR = 0.99; $P = 0.841$; 95% CI, (0.94, 1.06)) or individual-level perceived normative honour values (OR = 1.01; $P = 0.785$; 95% CI, (0.94, 1.09) (SPR); OR = 1.05; $P = 0.230$; 95% CI, (0.97, 1.14) (DFR); Supplementary Table 27) and coordinative efforts targeting achieving efficient coordination (that is, contributions of 6 MUs). However, the two facets of personal honour values showed divergent effects: SPR was related to a lower likelihood of contributing 6 MUs (OR = 0.88; $P = 0.001$; 95% CI, (0.82, 0.95)), while DFR was related to a higher likelihood of contributing 6 MUs (OR = 1.14; $P = 0.002$; 95% CI, (1.05, 1.23); Supplementary Table 27). These findings remained consistent when we used factor scores of honour values (Supplementary Table 28).

### Exploratory analyses: honour and behaviours adjusted by expectations
**(Less-)efficient coordination success.** To further shed light on the potential motives associated with the observed behavioural cooperation patterns, we compared the sum of individuals' own cooperation and expected partner's cooperation with two provision points of the public good. This allows us to explore how the cultural logic of honour relates to individuals' anticipation of coordination success (Fig. 3 and Methods). Societal-level honour positively predicted the anticipation

of efficient coordination success, defined as the expectation of reaching the higher provision point (OR = 1.42; $P < 0.001$; 95% CI, (1.26, 1.60)), but was not associated with the anticipation of less-efficient coordination success, defined as the expectation of reaching the lower but not the higher provision point (OR = 1.01; $P = 0.816$; 95% CI, (0.92, 1.11); Supplementary Table 29). At the individual level, perceiving stronger normative values of DFR was positively associated with anticipation of less-efficient coordination (OR = 1.20; $P < 0.001$; 95% CI, (1.10, 1.32)) but not with anticipation of efficient coordination (OR = 1.10; $P = 0.270$; 95% CI, (0.93, 1.29); Supplementary Table 29). The two facets of personal honour values showed divergent patterns: DFR positively predicted anticipation of efficient coordination success (OR = 1.19; $P = 0.030$; 95% CI, (1.02, 1.39)), while SPR negatively predicted anticipation of less-efficient coordination success (OR = 0.84; $P < 0.001$; 95% CI, (0.77, 0.91); Supplementary Table 29). The results were consistent when we used factor scores of honour values (Supplementary Table 30).

**(Less-)efficient competition.** We also explored different forms of competition by subtracting expected partner's competition from individuals' own competition. This allows to distinguish different types of competitive behaviour, which may reflect different underlying motives (Fig. 3 and Methods). Specifically, we explored how the cultural logic of honour relates to efficient competition (defined as spending just enough to win) and less-efficient competition (defined as overspending to make sure one wins). At the individual level, stronger perceived normative values of SPR consistently predicted a higher occurrence of efficient competition (OR = 1.11; $P = 0.012$; 95% CI, (1.02, 1.21)), but not less-efficient competition (OR = 0.97; $P = 0.497$; 95% CI, (0.88, 1.06); Supplementary Table 31). Perceived normative values of DFR did not predict the occurrence of either efficient or less-efficient competition (OR = 1.01; $P = 0.918$; 95% CI, (0.92, 1.10); Supplementary Table 31). These findings remained consistent when we used factor scores of honour values (Supplementary Table 32). However, we found no consistent evidence for an association between societal-level honour (or personal honour values) and the occurrence of either efficient or less-efficient competition using observed scores and factor scores of honour values (Supplementary Tables 31 and 32).

**(Un)conditional cooperation.** By subtracting expected partner's cooperation from individuals' own cooperation, we also distinguished different types of cooperative behaviour (Fig 3 and Methods) and explored how the cultural logic of honour relates to conditional cooperation (defined as matching the expected contribution of one's partner in the same round) and unconditional cooperation (defined as exceeding the expected contribution of one's partner in the same round). At the individual level, perceiving honour values of DFR as more prevalent in one's society consistently positively predicted the occurrence of conditional cooperation (OR = 1.10; $P = 0.043$; 95% CI, (1.00, 1.20)) but negatively predicted unconditional cooperation (OR = 0.82; $P < 0.001$; 95% CI, (0.73, 0.91); Supplementary Table 33). These findings were consistent when we used factor scores of honour values (Supplementary Table 34). However, we found no evidence for the association between societal-level honour (or individual-level honour indicators: perceived normative values of SPR and personal honour values for both facets) and the occurrence of either conditional or unconditional cooperation using observed scores and factor scores of honour values (Supplementary Tables 33 and 34).

## Discussion
Our online experiment tested hypotheses and research questions about the role of honour values in competition, cooperation and expectations of these behaviours from unrelated others, at both societal and individual levels, across 13 societies. The study incorporated a multi-faceted and multi-layered examination of honour values and norms, thereby providing a test of how the cultural logic of honour

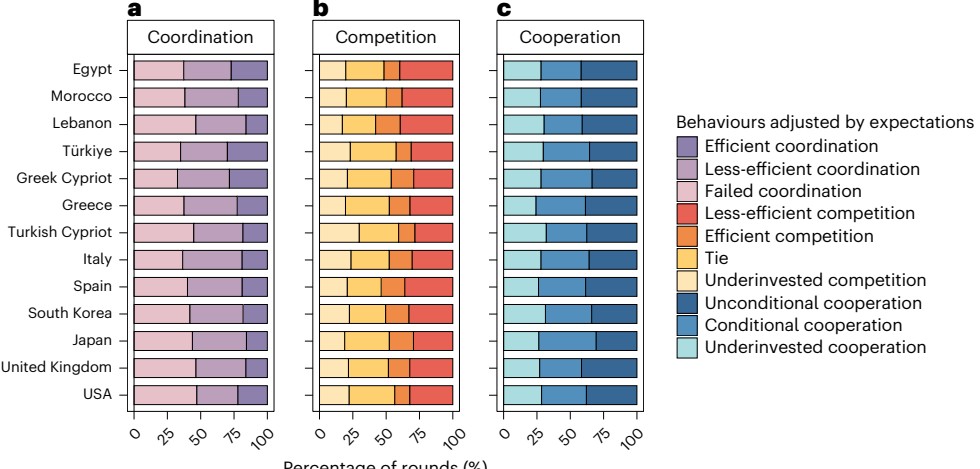

**Fig. 3 | Percentage of rounds for each type of anticipation of coordination success and behavioural deviation from expectations for competition and cooperation. a**, Anticipation of coordination success. The sum of an individual's own contribution and expected contribution from the other in a given round in the step-level PGG was grouped into three categories, where 'failed coordination' indicates that the sum contribution did not reach the first provision point (that is, 12 MUs), 'less-efficient coordination' indicates that the sum contribution only reached the first provision point but not the second one (that is, 16 MUs) and 'efficient coordination' indicates that the sum contribution reached the second provision point. **b**, Behavioural deviation from expectations for competition. In the contest game, the deviations of an individual's own competition from their expected competition from the other in a given round were grouped into four categories, where 'underinvested competition' indicates that the individual's own competition was less than the expected competition from the other, 'tie' indicates that the individual competed at exactly the same level as the expected

level from the other, 'efficient competition' indicates that the individual's own competition was just one MU more than the expected competition from the other and 'less-efficient competition' indicates that the individual's own competition was at least two MUs more than the expected competition from the other. **c**, Behavioural deviation from expectations for cooperation. In the step-level PGG, the deviations of an individual's own contribution from their expected contribution from the other in a given round were grouped into three categories, where 'underinvested cooperation' indicates that the individual's own contribution was less than the expected contribution from the other, 'conditional cooperation' indicates that the individual contributed exactly the same level as the expected level from the other and 'unconditional cooperation' indicates that the individual's own contribution was more than the expected contribution from the other. Societies are sorted in ascending order according to societal-level honour (that is, the societal mean of perceived normative honour values), from the bottom upwards on the *y* axis.

may shape competition and cooperation. As predicted, members of societies where honour values were more prevalent exhibited greater interpersonal competition (supporting H1a), but they did not show correspondingly higher expectations of competition from others in our main analyses (no support for H1b). Individuals who perceived honour values as more prevalent in their society also competed more (supporting H2a) and expected greater competition from others (supporting H2b). Personal honour values were not associated with competition (no support for H3). Similar patterns were observed for cooperation, with both societal mean and individual perceived normative honour values positively associated with cooperation and expectations of others' cooperation (see Table 2 for a summary of the main findings).

Our hypotheses and analyses were informed by the cultural logics framework, which conceptualizes honour as a cultural syndrome involving a set of coherent shared beliefs, values, behaviours and practices[26]. The positive association between perceived normative honour values and competition at both societal and individual levels aligns with characterizations of pre-emptive defence as an important strategy in social interactions under the cultural logic of honour[20,34,58–61], and with previous research on conflict and negotiation showing higher competitive aspirations in negotiations among individuals from honour cultural backgrounds than those from non-honour backgrounds[45]. Interestingly, exploratory analyses suggested that individuals who perceived stronger normative values of SPR may aim to minimize the cost of winning a contest rather than engage in excessive competitive spending that could diminish their welfare after winning. This finding challenges claims in the literature linking honour with abhorring cost–benefit calculations[26]. When competition is institutionalized with a clearly defined incentive structure, such conditions allow honour-related norms to manifest in efforts to compete efficiently, on the basis of expectations of the other's competition.

Beyond the conflict situation that constrained individuals to compete or not, the present study also employed a social coordination situation that afforded the possibility of working together to increase welfare. The positive association between perceived normative honour values and cooperation—including evidence from levels of cooperation, coordinative decisions targeting achieving efficient coordination (for example, contributing 8 MUs) and anticipation of coordination success—both at societal and individual levels, aligns with earlier research on honour cultures and conflict management. This research found that individuals from honour cultures, compared with those from non-honour cultures, were more willing and able to handle conflict situations constructively, and made more cooperative offers in negotiations when the situation afforded such opportunities—such as in the absence of insults[59] or in the presence of social rewards[49]. Moreover, exploratory analyses that subtracted expectations of others' cooperation from one's own suggested that individuals who perceived stronger normative values of DFR may be more likely to condition their own cooperation on the expected cooperation of others but less likely to respond altruistically to expected less-cooperative others. These findings provide empirical support for the theorized importance of positive reciprocal principles and self-protection to avoid being exploited in social interactions within the cultural logic of honour[26].

We observed a positive association between competition and cooperation at both the societal and individual levels, which supports the perspective that these two processes are not mutually exclusive but coexist[7,8]. Research has increasingly found competition and cooperation to co-occur for the same individuals in group activities[62] and across domains such as business[63] and politics[64]. Similarly, recent evolutionary models that investigated competition and cooperation as independent components have demonstrated the joint evolution of these behaviours[48]. Moreover, our findings suggest that competition and

**Table 2 | Support for hypotheses and summary of main findings**

| Predictor | Outcome | Competition | | | Cooperation |
| --- | --- | --- | --- | --- | --- |
| | | Hy. | Direction | Support | Direction |
| Societal-level honour | Behaviour | H1a | +* | Y | +* |
| | Expectation | H1b | + | N | +* |
| Individual-level honour | | | | | |
| Perceived normative honour values | | | | | |
| SPR | Behaviour | H2a | +* | Y | +** |
| DFR | | | +** | Y | + |
| SPR | Expectation | H2b | +* | Y | + |
| DFR | | | +** | Y | +*** |
| Personal honour values | | | | | |
| SPR | Behaviour | H3 | − | N | − |
| DFR | | | + | N | +** |
| Cross-level interactions | | | | | |
| Personal honour (SPR) × societal-level honour | Behaviour | | + | | +** |
| Personal honour (DFR) × societal-level honour | | | −* | | −* |
| Contextual effects | Behaviour | | + | | + |
| | Expectation | | − | | + |

Hy., hypothesis; Y, hypothesis supported; N, hypothesis not supported (non-significant results). Plus and minus signs indicate the direction of the effect. The contextual effects describe the differences in competition (or cooperation) among participants who have the same level of perceived normative and personal honour values but live in societies with different societal-level honour. There is no 'Support' column for cooperation as no hypothesis was preregistered. $*P < 0.05$; $**P < 0.01$; $***P < 0.001$

cooperation can coexist within the cultural logic of honour. This aligns with previous research that found self-reliance and group-oriented interdependence to coexist in societies where honour is a central cultural value[65] and to be associated with competition and cooperation[66,67]. Our findings suggest that the ecologies fostering the cultural logic of honour may also promote the co-emergence of competition and cooperation.

Our study provides multi-layered evidence by examining the cultural logic of honour from subjective endorsement of cultural values to intersubjective perceptions of normative values in one's society, and further extending to societal-level cultural phenomena[40,41,54,68]. Perceived normative honour values played a stronger and more robust role than personal values in predicting both individuals' behaviours and their expectations of others' behaviours in situations involving a conflict of interest. Aggregating these intersubjective perceptions to societal-level means as a cultural indicator largely replicated findings observed from individual-level perceived normative honour values. We further decomposed the societal-level effects into contextual and individual-level effects, but we found no evidence for contextual effects. This suggests that cultural contexts characterized by varying levels of honour value prevalence may shape interpersonal competition and cooperation primarily through individuals' perceptions of the prescribed values and norms within these contexts. Additionally, findings from cross-level interactions showed that personal honour values were more predictive of competition and cooperation in societies with lower societal-level honour. This suggests that weaker societal pressure to adhere to honour norms may amplify the role of personal honour values in shaping behaviours. Taken together, these findings highlight the importance of examining the cultural logic of honour as a set of normative values that individuals inhabiting different cultural contexts perceive and respond to, and of considering the affordances cultural contexts provide when testing the role of individual's personal beliefs or values in predicting their behaviours[41,69].

Our analyses revealed contrasting roles of two facets of personal honour values in relation to cooperation. Specifically, the value placed on DFR was associated with increased cooperative and coordinative efforts (the latter was particularly evidenced by more frequent decisions of equally splitting the cost to achieve successful coordination in the step-level PGG), whereas the value placed on SPR was linked to reduced efforts in the same behaviours. Divergent mechanisms also emerged for the two facets of honour when we examined the cross-level interactions in predicting cooperation. In societies with lower (versus higher) societal-level honour, personally endorsing SPR was found to hinder cooperation, while personally endorsing DFR played a positive role in fostering cooperation. One possible explanation lies in the interdependent and coordinative nature of family honour—a family's honour is maintained by members working together to uphold their family's reputation and prevent any damage to it in the surrounding environment[30]. However, it remains unclear why this family-honour-oriented coordination motive extended beyond close ingroup boundaries to also benefit unrelated others within the same society (in the absence of any outgroup from other societies). Future research could examine personal values of defending the honour of larger ingroups beyond the family to determine whether the same patterns hold at varying levels of group boundaries.

We used incentivized economic games to capture participants' actual behaviours (beyond hypothetical situations and questionnaire self-reports) as well as their incentivized expectations about others' behaviours. This approach introduces real consequences for individuals if their reported behaviour does not align with true preferences[39]. By altering the formal rules of the game, we applied structural variations to study specific types of situations[15]. For instance, the distinct separation between the contest game and the step-level PGG helped avoid ambiguity in operationalizing competitive and cooperative behaviours[7]. As evidenced by findings from reanalysis of previous datasets, step-level PGGs may be more suitable for measuring cooperation than PDs and continuous PGGs[13,16], as the strong appeal of non-cooperation to self-interest in the latter two may limit the expression of the cultural logic of honour in the manifestation of cooperation.

While past research has shown the ecological validity of behaviours measured in economic games[70–73], these insights may not generalize

to all social settings[74]. In everyday life, competition (and cooperation) involved in honour-claiming or honour-protecting behaviours may not adhere to formal rules or have an explicit incentive structure to determine winners and losers (determine provision points of public goods)[75]. Real-life cases of competition may sometimes result in mutual development rather than zero-sum outcomes[9]. Future research could employ methods such as experience sampling to explore the role of honour in shaping spontaneous competition and cooperation in daily social interactions. A further potential methodological limitation is that both competition and cooperation were measured as proactively deciding to invest resources. This approach may introduce confounds to the covariation of competition and cooperation with honour due to a general tendency among individuals to invest MUs into the (challenge/common) pool. However, this controlled for the potential framing effects that could arise if cooperation were operationalized as 'give-some' behaviour (that is, investing resources) and competition as 'keep-some' behaviour (that is, refraining from investing)[76,77].

The current research demonstrated a positive relationship between perceived normative honour values and competition, as well as cooperation, at both societal and individual levels across various societies. Personal values of DFR were linked to more cooperative and coordinative efforts, while SPR was associated with reduced efforts in these behaviours. These findings enhance our understanding of honour as a multi-faceted and multi-layered cultural logic shaping social interactions, particularly as individuals navigate conflict and coordination challenges with unrelated others in their society.

## Methods

### Ethics and inclusion
The research was approved by the Sciences & Technology Cross-Schools Research Ethics Committee at the University of Sussex (ER/SJ468/1). The preregistration (registered on 24 May 2023) and materials are accessible at https://osf.io/r9atc (see Supplementary Information section 1 for preregistration deviations and unregistered steps). All participants provided informed consent before voluntarily completing the study.

### Participants
We recruited 3,656 participants aged 18 years or older, stratified by age and gender, from 13 societies (Cyprus (both Greek and Turkish Cypriot communities), Egypt, Greece, Italy, Lebanon, Morocco, Spain, Turkey, Japan, South Korea, the United Kingdom and the USA). Several inclusion criteria were applied, resulting in the exclusion of 120 participants who were not born and located in the respective society, 24 participants who did not self-identify as male or female, 29 participants who failed the quality check question and 112 participants who failed all four comprehension questions designed to assess the participants' understanding of the contest game and step-level PGG rules. A final sample of 3,371 participants was retained for analyses (50.16% women; mean age, 40.79; s.d. of age, 14.36). Our sample was not stratified in terms of other demographic characteristics. The majority of participants self-identified as belonging to the majority ethnic group in the respective society (93.60%) and reported having an urban background (85.79%). Overall, participants reported a moderate level of parental education (that is, above high school; mean, 4.33; s.d., 1.58) and subjective socio-economic status (mean, 5.59; s.d.,1.92; on a scale from 1 to 10; see Supplementary Table 35 for more information). One of our main goals was to detect potential differences between societies in their levels of competition and cooperation. A sensitivity power analysis indicated that a sample of 250 participants per society, with 80% power ($\alpha = 0.05$), could detect an effect size of $d = 0.25$ between two societies. We thus aimed at recruiting 3,250 participants (~250 per society). Participants were recruited through an online panel provider (Toluna), including members of its third-party panel providers. As an exception, participants from Cyprus were recruited through a market

research agency based in the Greek Cypriot community (CYMAR) and a research, analysis and consultancy organization based in the Turkish Cypriot community (Statica). Participants either received an email invitation or had access to the study link through the panellist portals. Only participants in the Turkish Cypriot community completed the study on a tablet provided by the research organization. The participants were compensated for their participation right after completing the survey and received additional payment based on their own and their paired game partner's decisions at the end of data collection in each society.

### Procedure and experimental design
The design consisted of two counter-balanced within-participant treatments with type of game (that is, contest game and step-level PGG) and three randomized within-participant treatments related to the gender information of the pairing partner (that is, male versus female versus gender not provided). We collected data using the software platform Qualtrics (version May 2023). The study materials were prepared in English and translated into local languages of the non-English-speaking countries following a team translation approach[77,78]. Specifically, all materials were first translated by members of the research team who are native speakers of the respective language, and then reviewed and checked for accuracy and local conventions of language use by other team members who are fluent in both the local language and English. Whenever disagreements emerged, an additional round of discussion was used to reach a final decision. In some cases, we adjusted the wording of materials to fit locally common expressions (for example, the translation of 'challenge pool' for the contest game).

The same experimental procedure was followed in all samples. The participants were asked to make six independent rounds of decisions in the contest game and another six rounds in the step-level PGG. Each round involved a different game partner—male, female or with gender information not provided—from their own society, whose decisions were asynchronously paired with those of the participant after the experiment. The participants were asked to make decisions regarding the allocation of MUs and estimate their partners' decisions. To ensure comparable payment levels, each MU was set to the monetary value of 0.1 kg of flour in each society. Information on flour prices in each society was retrieved at https://www.globalproductprices.com/ in March 2023. The participants were informed about the monetary value of each MU and that their decisions in the game would have monetary consequences. No deception was used in the economic games. The participants also completed several measures, including perceived normative values and personal values across the two facets of honour (that is, SPR and DFR), beliefs in a zero-sum game and relational mobility. They were debriefed at the end of the experiment and compensated for their participation through the panel provider or research agency.

After data collection was completed, we randomly selected one of 12 rounds of participants' decisions from the two economic games for post hoc decision pairing within each society and calculating participants' payment from the games[16,79]. The pairing of decisions was implemented on the basis of both the participant's gender and the partner's gender information from the randomly selected round. For example, if a female participant's game partner in the selected round was male, her decision was paired with that of a male participant whose game partner was female. The game payment consisted of earnings from making the decision and from making an accurate estimation of their partner's decision in the selected round. The participants received their game payment within two weeks following the conclusion of data collection.

### Contest game
We applied a continuous contest game (also referred to as the rent-seeking game)[43,80,81] to measure individuals' own competitive behaviour and expectations of others' competition. The contest game involved two players. Each player received an endowment of 10

MUs and decided how many of the 10 MUs they wanted to invest into a challenge pool ($x_i$ denotes player $i$'s investment; $0 \leq x_i \leq 10$) or keep for themselves. Higher investment in the challenge pool was taken as evidence of individuals engaging in higher levels of competitive behaviours. The player who invested more in the challenge pool won the game and received final earnings comprising the remaining MUs that the other player did not invest plus the MUs that the winning player kept for themselves. In other words, the winner of the game took the remaining resources of the loser, and the loser ended up with nothing. However, if the two players invested equal MUs to the challenge pool (that is, tied), both players simply ended up with the MUs they did not invest in the challenge pool. More formally, if $\pi_i$ denotes player $i$'s pay-off, then

$$\pi_i = \begin{cases} (10 - x_i) + (10 - x_j), & \text{if } x_i > x_j \text{ (that is, } i \text{ wins)} \\ 10 - x_i, & \text{if } x_i = x_j \text{ (that is, } i \text{ ties)} \\ 0, & \text{if } x_i < x_j \text{ (that is, } i \text{ loses).} \end{cases}$$

The contest game is thus a symmetric conflict game in which each player has the possibility to increase their pay-off at the expense of the other player. In this game, player $i$'s pay-off would fall in the range of $0 \leq \pi_i \leq 19$ MUs. The Pareto efficient outcome could be achieved if no player invested to exploit the other and both kept their initial endowment (and thereby maintained peace). However, peace is game-theoretically unstable since there is always a temptation for one of the players to invest just one MU to the challenge pool and thereby take all the MUs of the other player in this case (see Supplementary Information section 5.1 for more information).

## Step-level PGG

We applied a step-level PGG to measure cooperation and coordination[5,53]. This step-level PGG involved two players and two provision points. Each player received an endowment of 10 MUs and decided how many of the 10 MUs they wanted to invest into a common pool ($0 \leq x_i \leq 10$) or keep for themselves. Higher investment in the common pool was taken as individuals engaging in higher levels of cooperative behaviour. Both players' investment in the common pool would be lost if the total investment did not reach the first provision point of 12 MUs. If the total investment reached 12 MUs, each player received 10 MUs from the common pool. Moreover, if the total investment reached the second provision point of 16 MUs, each player received 15 MUs from the common pool. More formally:

$$\pi_i = \begin{cases} 10 - x_i, & \text{if } x_i + x_j < 12 \\ 10 - x_i + 10, & \text{if } 12 \leq x_i + x_j < 16 \\ 10 - x_i + 15, & \text{if } 16 \leq x_i + x_j. \end{cases}$$

The implementation of two provision points allowed the step-level PGG to have coordinated solutions—that is, players could work together to increase their pay-off through successful coordination. Player $i$'s pay-off would fall in the range of $0 \leq \pi_i \leq 19$ MUs. We defined successful coordination as cases without wasteful investment (that is, cases where $x_i + x_j \in \{0, 12, 16\}$) and efficient coordination as the case when the provision of the public good maximized joint pay-offs (that is, $x_i + x_j = 16$). Players had an incentive to make higher contributions, as efficient coordination always yielded higher pay-offs than less efficient coordination (that is, $x_i + x_j = 12$). However, it was not safe for individuals to invest in the common pool, because the first provision point of 12 MUs could not be exceeded alone, and the second provision point of 16 MUs required high investment from both players. One could waste one's own investment if the other player did not make a sufficient investment (see Supplementary Information section 5.2 for more information).

## Expectations about others' competition and cooperation

After each competition or cooperation decision, the participants were asked about their expectation of their partner's behaviour (on a scale of 0 to 10). We incentivized these expectations using a simple belief elicitation rule[81]. Specifically, participants earned 5 MUs if they made a correct estimation of their partner's behaviour. Participants' pay-off from making an estimation $\pi_e$ equalled 5 when the estimation was correct or 0 when it was incorrect.

## Behaviours adjusted by expectations

In the step-level PGG, we also distinguished different types of anticipation of coordination success by summing up an individual's cooperation and their expectations of their game partner's cooperation. Specifically, we categorized a given round as efficient coordination if the expected sum contribution reached the second provision point (that is, 16 MUs or more), as less-efficient coordination if it only reached the first provision point (that is, 12 MUs or more but fewer than 16 MUs) and otherwise as failed coordination (that is, fewer than 12 MUs; Fig. 3).

In the contest game, we distinguished different types of competition by analysing behavioural deviation from expectations—that is, subtracting individuals' expectations of their game partners' competition from their own competition decisions. Specifically, a given round can be categorized as underinvested competition if the deviation of an individual's competition from the expected competition of the opponent was negative (meaning that they anticipated losing their money), as a tie if the deviation was equal to zero MU, as efficient competition if the deviation was equal to one MU (because an individual could potentially win the contest game with minimal investment, thereby retaining the most remaining resources) and as less-efficient competition if the deviation was higher than one MU (because any positive deviations greater than one might ensure a win but reduced the individual's overall pay-off in that round; Fig. 3).

In the step-level PGG, we distinguished different types of cooperation by analysing behavioural deviation from expectations—that is, subtracting individuals' expectations of their game partners' cooperation from their own cooperation decisions. Specifically, we categorized a given round as underinvested cooperation if the deviation of an individual's own cooperation from the expected cooperation of the game partner was negative (meaning that they anticipated contributing less than their partner), as conditional cooperation if the deviation was zero MU (because an individual anticipated that their own level of cooperation would match their partner's cooperation in that round) and as unconditional cooperation if the deviation was positive (because an individual anticipated contributing more than their partner, rather than matching their contributions with their partner's level of cooperation; Fig. 3).

## Honour values

The participants were asked to rate ten items assessing their endorsement of two facets of honour values: DFR (for example, 'People should not allow others to insult their family') and SPR (for example, 'People always need to show off their power in front of their competitors')[21,54]. The participants rated the same set of items twice: once indicating their personal honour values ('How much do you agree or disagree with the following statements?') and another time indicating their perceived normative honour values—that is, their perception of the extent to which most people in their society would agree or disagree with the items ('How much would most people in your society agree or disagree with the following statements?'). The order of these two ratings was counterbalanced across participants. Responses to items were given on a seven-point scale (1 (strongly disagree) to 7 (strongly agree) for personal endorsement; 1 (most people would strongly disagree) to 7 (most people would strongly agree) for societal perception). Higher scores indicate stronger personal honour values or perceived normative honour values.

## Beliefs in a zero-sum game

Beliefs in a zero-sum game capture the generalized beliefs about the nature of social relations involving completely conflicting interests[55]. Previous research has shown that these beliefs can lead to competition and conflict, and they vary across societies and social economic status[55,82]. To examine whether beliefs in a zero-sum game explained additional variation in competition beyond what could be explained by honour values, we measured this construct by asking the participants to indicate the extent to which they agreed with eight statements about their belief that life is conceived as a zero-sum game (for example, 'The successes of some people are usually the failures of others'; 1 (strongly disagree) to 6 (strongly agree)). Higher scores indicate stronger beliefs in a zero-sum game.

## Relational mobility

Relational mobility is a socio-ecological variable that represents how much freedom and opportunity a society affords individuals to choose and dispose of interpersonal relationships according to personal preference[56]. Past research has found higher levels of cooperation in societies characterized by more flexible and fluid social relations, as well as among individuals who perceive their environment as offering more opportunities to establish new relationships with strangers[16]. To examine whether relational mobility explained additional variation in cooperation beyond what could be explained by honour values, we measured this variable by asking the participants to state how well 12 statements described the people in the society where they lived (for example, 'It is common for these people to have a conversation with someone they have never met before'; 1 (strongly disagree) to 6 (strongly agree)). Higher scores indicate that people perceive their society to promote open and flexible social relations.

## Demographic information

The participants were also asked to indicate their age, gender, country of birth, length of stay in the country of data collection, type of environment they mainly lived in (urban, rural or both), ethnic background, religious background, religiosity, education level of their parents and their own subjective social status in their country of residence[83]. All demographic materials were adjusted to the respective country by local collaborators, ensuring that the questions assessed locally meaningful categories (for example, the category of religious background varies across countries).

## Other societal-level indicators

The cultural logic of honour has been argued to emerge in harsh, competitive environments characterized by high status inequality and mobility and by historically weak institutions[27–29]. To operationalize the characteristics of these environments, we selected a set of theoretically relevant societal-level indicators that were retrievable for as many societies in the current study as possible. These included economic indicators (gross domestic product per capita, gross national income per capita, human development index and gender inequality), quality of institutions indicators (government effectiveness, rule of law, stability/violence, corruption control, corruption perceptions index and market competitiveness), and historical and ecological threats (historical prevalence of infectious disease, world risk index, exposure and vulnerability). Except for the Turkish Cypriot community, these indicators were available for all societies in the current study (see Supplementary Table 13 for more information about the operationalization of these societal-level indicators).

## Analytic strategy

For the societal-level hypotheses (H1a and H1b), we applied mixed-effects models in which participants (level 2) and societies (level 3) were included as two random intercepts, and we tested societal-level honour as a fixed predictor. For the individual-level hypotheses (H2a, H2b and H3), we applied mixed-effects models in which participants (level 2) and societies (level 3) were included as two random intercepts to test whether perceived normative values and personal values of honour relate to competition, cooperation or expectations of these behaviours from others. We calculated separate indicators of each facet of perceived normative honour values as well as of personal honour values, and simultaneously included all four individual-level honour indicators as predictors in the mixed-effects model. This approach allowed us to test the roles of perceived normative values and personal values while controlling for each one, as well as to examine how each facet uniquely explained variation in behaviours and expectations. As preregistered, age and participant gender were entered in these models as control variables. We also preregistered the inclusion of the number (that is, order) of the randomized game rounds as a control, but we were unable to retrieve this information from the Qualtrics survey due to programming constraints. To address this limitation, we instead included the order of the game and gender information of the pairing partner as additional control variables (Supplementary Table 1). Gender information of the pairing partner and order of the game were level 1 controls in the models. Individual differences variables (age and participant gender) were level 2 controls. We analysed the data with R 4.2.1 (ref. 84) (lme4 package[85] v.1.1-35.5). All significance tests were two-tailed.

For multi-item measures of individual-level honour indicators, beliefs in a zero-sum game and relational mobility, we used observed scores, calculated as unweighted means of the respective scale items. We also generated a societal-level indicator of honour based on mean perceived normative honour values across the two facets for each society, as well as societal-level indicators of beliefs in a zero-sum game and relational mobility based on the societal means of these variables. To ensure the robustness of our analyses, we also obtained factor scores for honour values at both the between-society and within-society levels using confirmatory factor analysis and adjusting for response styles in Mplus 8.10 (ref. 86) (see Supplementary Information section 2 for more information). Additional analytic strategies used for robustness checks and exploratory purposes are detailed in the Supplementary Information.

## Reporting summary

Further information on research design is available in the Nature Portfolio Reporting Summary linked to this article.

## Data availability

The datasets generated and analysed during the current study are publicly available via OSF at https://osf.io/3dscw/.

## Code availability

The code used to analyse the data is publicly available via OSF at https://osf.io/3dscw/. The R code is also provided on the Code Ocean platform (https://doi.org/10.24433/CO.9371203.v1), allowing for a straightforward reproducible run.

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

## Acknowledgements

We thank J. O'Brien, M. Tahsin, S. Syed, H. Jun and J. Song for their help in different stages of the research. This research was funded by a European Research Council Consolidator Grant (HONORLOGIC, no. 817577) awarded to A.K.U. The funder had no role in study design, data collection and analysis, decision to publish or preparation of the manuscript.

## Author contributions

S.J., A.K.U., A.R. and V.L.V. conceived the project, designed the study and discussed the results. S.J. implemented the study with translation support from R.R.-B., M.G.Y., C.H., S.H., K.I., P.K., K.K., E.K., J.M.-L., R.M., J.N., S.P., C.P., D.R., M.T., Y.U. and A.K.U.; analysed data with input from V.L.V. and A.K.-H.; and wrote the paper with input from A.K.U., A.R., V.L.V., A.K.-H., R.R.-B., S.E.C., M.G.Y., C.H., S.H., K.I., P.K., K.K., E.K., J.M.-L., R.M., J.N., S.P., C.P., D.R., M.T. and Y.U. A.K.U. supervised the project.

## Competing interests

The authors declare no competing interests.

## Additional information

**Correspondence and requests for materials** should be addressed to Shuxian Jin or Ayse K. Uskul.

¹School of Psychology, University of Sussex, Brighton, UK. ²Social, Economic and Organisational Psychology Department, Leiden University, Leiden, the Netherlands. ³Institute for the Future of Human Society, Kyoto University, Kyoto, Japan. ⁴Mind, Brain, and Behaviour Research Center (CIMCYC), University of Granada, Granada, Spain. ⁵Department of Psychology, Iowa State University, Ames, IA, USA. ⁶Institute for Interdisciplinary Research on Conflict and Violence, Bielefeld University, Bielefeld, Germany. ⁷Psychology Program, Doha Institute for Graduate Studies, Doha, Qatar. ⁸Department of Psychology, Eastern Mediterranean University, Famagusta, Cyprus. ⁹Department of Cognitive and Psychological Sciences, Nagoya University, Nagoya, Japan. ¹⁰Department of Psychology, University of Crete, Rethymno, Greece. ¹¹School of Psychology, Aristotle University of Thessaloniki, Thessaloniki, Greece. ¹²Department of Psychology, Palacký University Olomouc, Olomouc, Czech Republic. ¹³Department of Social and Organizational Psychology, Faculty of Psychology, National University of Distance Education, Madrid, Spain. ¹⁴Department of Economics, The British University in Egypt, Cairo, Egypt. ¹⁵Department of Psychology, Sogang University, Seoul, South Korea. ¹⁶Department of Psychology, University of Chieti-Pescara, Chieti, Italy. ¹⁷Department of Psychology, University of Cyprus, Nicosia, Cyprus. ¹⁸Faculty of Social Sciences, Northeastern University London, London, UK. ¹⁹Department of Education, Cultural Heritage and Tourism, University of Macerata, Macerata, Italy. ²⁰These authors contributed equally: Angelo Romano and Vivian L. Vignoles. ✉e-mail: shuxian.jin@sussex.ac.uk; A.K.Uskul@sussex.ac.uk

# Reporting Summary

## Statistics

For all statistical analyses, confirm that the following items are present in the figure legend, table legend, main text, or Methods section.

| n/a | Confirmed | |
|---|---|---|
| ☐ | ☒ | The exact sample size (*n*) for each experimental group/condition, given as a discrete number and unit of measurement |
| ☐ | ☒ | A statement on whether measurements were taken from distinct samples or whether the same sample was measured repeatedly |
| ☐ | ☒ | The statistical test(s) used AND whether they are one- or two-sided<br>*Only common tests should be described solely by name; describe more complex techniques in the Methods section.* |
| ☐ | ☒ | A description of all covariates tested |
| ☒ | ☐ | A description of any assumptions or corrections, such as tests of normality and adjustment for multiple comparisons |
| ☐ | ☒ | A full description of the statistical parameters including central tendency (e.g. means) or other basic estimates (e.g. regression coefficient) AND variation (e.g. standard deviation) or associated estimates of uncertainty (e.g. confidence intervals) |
| ☐ | ☒ | For null hypothesis testing, the test statistic (e.g. *F*, *t*, *r*) with confidence intervals, effect sizes, degrees of freedom and *P* value noted<br>*Give P values as exact values whenever suitable.* |
| ☒ | ☐ | For Bayesian analysis, information on the choice of priors and Markov chain Monte Carlo settings |
| ☐ | ☒ | For hierarchical and complex designs, identification of the appropriate level for tests and full reporting of outcomes |
| ☐ | ☒ | Estimates of effect sizes (e.g. Cohen's *d*, Pearson's *r*), indicating how they were calculated |

*Our web collection on statistics for biologists contains articles on many of the points above.*

## Software and code

Policy information about availability of computer code

| Data collection | Individual responses were collected using the Qualtrics software (Version May 2023). |
|---|---|
| Data analysis | Data were analyzed using the software R (version 4.2.1). Confirmatory factor analyses were conducted using the software Mplus (version 8.10). The code used to analyse the data is publicly available at https://osf.io/3dscw/. The R code is also provided on the Code Ocean platform (https://doi.org/10.24433/CO.9371203.v1), allowing for a straightforward reproducible run. |

For manuscripts utilizing custom algorithms or software that are central to the research but not yet described in published literature, software must be made available to editors and reviewers. We strongly encourage code deposition in a community repository (e.g. GitHub). See the Nature Portfolio guidelines for submitting code & software for further information.

## Data

Policy information about availability of data

All manuscripts must include a data availability statement. This statement should provide the following information, where applicable:
- Accession codes, unique identifiers, or web links for publicly available datasets
- A description of any restrictions on data availability
- For clinical datasets or third party data, please ensure that the statement adheres to our policy

The datasets generated and analysed during the current study are publicly available at https://osf.io/3dscw/.

# Research involving human participants, their data, or biological material

Policy information about studies with human participants or human data. See also policy information about sex, gender (identity/presentation), and sexual orientation and race, ethnicity and racism.

| | |
|---|---|
| Reporting on sex and gender | Participants were asked to self-report their sex/gender (male; female; not listed, please specify) at the beginning of the study, following the provision of informed consent. Only those who self-identified as male or female were included in the data analysis. In surveys conducted in the Arabic (Egypt, Lebanon, Morocco), Greek (Greece, Greek Cypriot community), Japanese (Japan), Korean (South Korea), and Turkish (Türkiye, Turkish Cypriot community) languages, the terms "sex" and "gender" were translated using the same word. In the English (U.K., U.S.A.), Italian (Italy), and Spanish (Spain) versions of the survey, (the translation of) the term "sex" was used in the question. <br><br> Across these 13 societies, we consider this self-reported measure closer to the working definition of gender rather than a strictly binary concept of sex, provided on the Nature portfolio, because participants were always provided a third option (i.e., not listed, please specify) to indicate self-identifications beyond male and female. <br><br> We also manipulated the sex/gender information of the game partner (male vs. female vs. not provided) with whom the participants' decision would be paired if that game round was selected for game payment calculation. Across these 13 societies, we consider this manipulation to reflect partner gender information for the same reasons mentioned above. <br><br> Individual-level participant gender information is provided in the source data. Consent has been obtained for sharing de-identified individual-level data. |
| Reporting on race, ethnicity, or other socially relevant groupings | Participants were asked to self-report their ethnic and religious backgrounds at the end of the study. The categories provided for these demographic questions were adapted to each society by local collaborators to ensure that they reflected locally meaningful categories. Individual-level data on these variables are provided in the source data. In Table S35 of the Supplementary Information, we report the percentage of participants who self-identified as belonging to the major ethnic group in the respective society (see the column "% Majority"). These two variables were not used as control variables in the analysis. |
| Population characteristics | See "Behavioural & social sciences study design" section. |
| Recruitment | Participants were recruited through an online panel provider (Toluna, https://www.toluna.com/) including members of its third-party panel providers. As an exception, participants from Cyprus were recruited through a market research agency based in the Greek Cypriot community (CYMAR, https://www.cymar.com.cy/), and a research, analysis and consultancy organization based in the Turkish Cypriot community (Statica, https://staticacy.com/). See Table S35 of the Supplementary Information for more details about the panels. Participants in all 13 societies were compensated for their participation in the study, and also received additional payment based on their own and their paired game partner's decisions at the end of data collection in each society. To minimize self-selection bias, we did not set specific requirements for participation. The recruitment template included only general information about the estimated survey length and the compensation for participation. |
| Ethics oversight | This study was approved by the Sciences & Technology Cross-Schools Research Ethics Committee (C-REC) at the University of Sussex (ER/SJ468/1). |

Note that full information on the approval of the study protocol must also be provided in the manuscript.

# Field-specific reporting

Please select the one below that is the best fit for your research. If you are not sure, read the appropriate sections before making your selection.

☐ Life sciences ☒ Behavioural & social sciences ☐ Ecological, evolutionary & environmental sciences

For a reference copy of the document with all sections, see nature.com/documents/nr-reporting-summary-flat.pdf

# Behavioural & social sciences study design

All studies must disclose on these points even when the disclosure is negative.

| | |
|---|---|
| Study description | Quantitative data, experimental and correlational design |
| Research sample | This study involved participants from 13 societies, recruited from participant pools provided by panel providers, including Toluna and its third-party partners, CYMAR and Statica. The percentage of females in the final sample ranged from 48% to 53%, and participants' mean age ranged from 39.25 (SD = 12.83) to 41.56 (SD = 14.91) across the 13 societies. These panel providers were chosen because their samples are heterogeneous in terms of age, gender, and socio-economic background. Due to limited access to participants in the Greek and Turkish Cypriot communities through Toluna and its third-party partners, we collaborated with local research companies for data collection in these communities. |
| Sampling strategy | Participants were recruited from participant pools of the panel providers in each society. We stratified the participants by age and gender in each participant pool by setting quota groups for age (18-25, 26-35, 36-45, 46-55, 56+) and gender (male and female) at |

the beginning of the Qualtrics surveys.

One of our main goals was to detect potential differences between societies in their level of competition and cooperation. A sensitivity power analysis indicated that a sample of 250 participants per society, with 80% power ($\alpha$ = .05), could detect an effect size of d = .25 between two societies. We thus aimed at recruiting 3,250 participants (~250 per society).

**Data collection**

This study involved anonymized online data collection. Participants either received an email invitation or accessed the Qualtrics survey link through panelist portals. Only participants in the Turkish Cypriot community completed the Qualtrics survey on a tablet provided by the research organization, in a separate room and alone, without the presence of the research representative who was blind to the study hypotheses and experimental conditions. Thus, across all societies, researchers could not influence the results knowing the hypotheses and the experimental conditions in advance.

**Timing**

Data collection in the Greek Cypriot community, managed by CYMAR: June 1, 2023 to June 9, 2023
Data collection in the Turkish Cypriot community, managed by Statica: June 21, 2023 to October 21, 2023
Data collection in the rest of the societies, managed by Toluna and its third-party panel provider: May 23, 2023 to June 13, 2023

**Data exclusions**

Several inclusion criteria were applied, resulting in the exclusion of a) 120 participants who were not born and located in the respective society, b) 24 participants who did not self-identify as male or female, c) 29 participants who failed the quality check question, and d) 112 participants who failed all four comprehension questions designed to assess participants' understanding of the contest game and step-level PGG rules. These criteria were established in consultation with panel providers regarding the availability of eligible samples in their participant pools before and during the data collection stage.

**Non-participation**

The response rate for each society, calculated as the final sample divided by the number of participants who agreed to participate and passed the quota group checks, was as follows: Egypt (75%), Greece (67%), Greek Cypriot community (80%), Italy (68%), Japan (54%), South Korea (71%), Lebanon (83%), Morocco (72%), Spain (47%), Türkiye (60%), Turkish Cypriot community (79%), United Kingdom (50%), and United States (33%).

**Randomization**

The design consisted of two counter-balanced within-participant treatments with type of game (i.e., contest game, step-level public goods game) and three randomized within-participant treatments related to the gender information of the pairing partner (i.e., male vs. female vs. gender information not provided). Thus, participants were not allocated into experimental groups. Counter-balancing and randomization were handled by Qualtrics. Both the order of the game and partner gender information were included in the analyses as control variables.

# Reporting for specific materials, systems and methods

We require information from authors about some types of materials, experimental systems and methods used in many studies. Here, indicate whether each material, system or method listed is relevant to your study. If you are not sure if a list item applies to your research, read the appropriate section before selecting a response.

## Materials & experimental systems

| n/a | Involved in the study |
|-----|----------------------|
| ☒ | ☐ Antibodies |
| ☒ | ☐ Eukaryotic cell lines |
| ☒ | ☐ Palaeontology and archaeology |
| ☒ | ☐ Animals and other organisms |
| ☒ | ☐ Clinical data |
| ☒ | ☐ Dual use research of concern |
| ☒ | ☐ Plants |

## Methods

| n/a | Involved in the study |
|-----|----------------------|
| ☒ | ☐ ChIP-seq |
| ☒ | ☐ Flow cytometry |
| ☒ | ☐ MRI-based neuroimaging |

## Plants

**Seed stocks**

*Report on the source of all seed stocks or other plant material used. If applicable, state the seed stock centre and catalogue number. If plant specimens were collected from the field, describe the collection location, date and sampling procedures.*

**Novel plant genotypes**

*Describe the methods by which all novel plant genotypes were produced. This includes those generated by transgenic approaches, gene editing, chemical/radiation-based mutagenesis and hybridization. For transgenic lines, describe the transformation method, the number of independent lines analyzed and the generation upon which experiments were performed. For gene-edited lines, describe the editor used, the endogenous sequence targeted for editing, the targeting guide RNA sequence (if applicable) and how the editor was applied.*

**Authentication**

*Describe any authentication procedures for each seed stock used or novel genotype generated. Describe any experiments used to assess the effect of a mutation and, where applicable, how potential secondary effects (e.g. second site T-DNA insertions, mosiacism, off-target gene editing) were examined.*

