## [Peer Review File · Nature Human Behaviour]

Honour, Competition and Cooperation across 13 Societies

Corresponding Author: Dr Shuxian Jin

Version 0:

Decision Letter:

20th January 2025

Dear Dr Jin,

Thank you once again for your manuscript, entitled "Honor, Competition and Cooperation across 13 Societies," and for your patience during the peer review process.

Your manuscript has now been evaluated by 2 reviewers, whose comments are included at the end of this letter. Although the reviewers find your work to be of interest, they also raise some important concerns. We are very interested in the possibility of publishing your study in Nature Human Behaviour, but would like to consider your response to these concerns in the form of a revised manuscript before we make a decision on publication.

To guide the scope of the revisions, the editors discuss the referee reports in detail within the team, including with the chief editor, with a view to (1) identifying key priorities that should be addressed in revision and (2) overruling referee requests that are deemed beyond the scope of the current study. We hope that you will find the prioritised set of referee points to be useful when revising your study. Please do not hesitate to get in touch if you would like to discuss these issues further.

1. Our reviewers raise concerns about the conceptualisation of core concepts. In your revisions, please ensure that you use literature-informed clear conceptualisation of the investigated concepts.
2. Both reviewers have concerns about the presentation of your findings. While Reviewer 1 finds the presentation to be quite descriptive and lacking a broader framework, Reviewer 2 finds that the main message is diluted because of the many analyses and findings. When revising your manuscript, please ensure that you streamline the presentation of the results and findings and place them in a broader theoretical context. We ask that you do not remove findings from the manuscript, especially not when they were pre-registered.
3. Reviewer 1 requests that you provide additional methodological details and provide a rationale for the inclusion of individual variables. Please address these requests in full.

In sum, we invite you to revise your manuscript taking into account all reviewer and editor comments. We are committed to providing a fair and constructive peer-review process. Do not hesitate to contact us if there are specific requests from the reviewers that you believe are technically impossible or unlikely to yield a meaningful outcome.

We hope to receive your revised manuscript within two months. I would be grateful if you could contact us as soon as possible if you foresee difficulties with meeting this target resubmission date.

- Include a "Response to the editors and reviewers" document detailing, point-by-point, how you addressed each editor and referee comment. If no action was taken to address a point, you must provide a compelling argument. When formatting this document, please respond to each reviewer comment individually, including the full text of the reviewer comment verbatim followed by your response to the individual point. This response will be used by the editors to evaluate your revision and sent back to the reviewers along with the revised manuscript.

- Highlight all changes made to your manuscript or provide us with a version that tracks changes.

- **EXTENDED DATA FIGURES**

Link Redacted

We look forward to seeing the revised manuscript and thank you for the opportunity to review your work. Please do not hesitate to contact me if you have any questions or would like to discuss these revisions further.

Sincerely,

[Redacted]

[Redacted]

Nature Human Behaviour

Reviewer expertise:

Reviewer #1: honour ; competition/cooperation ; cross-cultural psychology

Reviewer #2: honour ; competition/cooperation ; cross-cultural psychology

REVIEWER COMMENTS:

Reviewer #1 (Remarks to the Author):

Review

This study has the potential to contribute to the deeper understanding of the psychological components of honor culture. The study is a very complex design mainly focusing on the relationship of competitive and cooperative behavioral choices in experimental conditions with perceived normative and personal honor values. All this is done in 12 countries providing opportunity to highlight similarities and differences among cultures endorsing different honor norms. The study has a number of novel results that are worth to be made available.

The experimental design and the statistical analysis are very complex and adequately serve the purpose of the research goals. My review focuses primarily on the theoretical part and the discussion part because the introductory part and concluding part in my view should be way more elaborated to highlight the relevance of the study and its potential contribution to deeper understanding of the psychology of honor, its relationship with competition and cooperation and in different cultures.

Theoretical framework

Concepts: competition, contest, conflict

- It would be useful to differentiate between competition and contest. Competition includes both formal/structured (contests) and informal/spontaneous (social comparison based) competitions (Sommer, 1995). Contests are typically institutionalized/formal competitions (e.g. sports, tournaments, beauty contests, presidential election, applying for management position) in which there are clear criteria and several contestants and limited resources/prizes. Please use the right expressions. This is important because the competition that is involved in honor protecting behavior is not an institutionalized contest, does not have formal rules, clear criteria who is the winner and loser and clear reward system but it is an informal competitive behavior involving many potentially different self-chosen ways to achieve the restoration of the honor of the person or the family. Maybe the word "contest" is valid for the experimental situation involved in this study, but competition is the right word for the interpersonal dynamic that takes place in honor protection. The experimental design is a kind of contest, but honor protection related competition is an informal competition and is not a zero-sum game. If one'a

honor is protected that does not mean that the other party will have less, or in other words one's gain is not one's cost necessarily.

- Also in the text the authors use competition and conflict interchangeably. "The present study used incentivized economic games to differentiate social interactions in conflict and cooperation situation". This indicates that they endorse the a concept that competition is always a conflict (zero-sum game) which is not the case, take for example self-developmental competitiveness (Ryckman et al, 1996; Orosz et al, 2018) and mutual development competitiveness (Fülöp, 2004 , Fülöp, 2009).

Competition and cooperation

- The study has an ongoing theoretical ambiguity if competition and cooperation should be handled together, if they are opposites or not and it quite apparent that the authors are much more familiar with the literature on honor than on the literature and state of the art of competition research. First of all, the fact that they study both competition and cooperation is not fully explained because they provide a short theoretical explanation why honor culture and competition can be related, but they cannot provide a convincing theoretical framework why cooperation should be studied also in relation with honor. It seems they think if competition is related than cooperation also has to. They seem to endorse to a degree the already overcome concept that competition and cooperation are opposites because, they realize with "surprise" in their study that they are not.
 - In the introduction they write for example: "However, recent findings suggest that a lack of cooperation does not equate to the presence of competition" But this is not recent. The fact that competition and cooperation are not dichotomous and polar opposites have been present in the literature since the 1990's. Even Morton Deutsch (1990), who was mainly the source of competition and cooperation being "dichotomous" wrote that they are rarely found in their pure form in real life, because for example most conflicts are a mixture of competing and cooperating processes, and the course and consequences of the conflict depend strongly on the nature of this competing-cooperating mixture. Since the 1990s there is an abundant literature on this (e.g. Van de Vliert, 1997) so being "surprised" is for the newcomer in this field . The literature should be important to be familiar with also with the literature on for example cooperative competition (e.g. Fülöp & Takács, 2013). This would also help the authors provide an explanation why it is worth to study how honour as a "cultural logic" affects cooperation. There is not real explanation why cooperation was included – what is its potential psychological/behavioural relationship with honor. It seems that the main idea was to study competition and because formerly the literature tended to handle cooperation and competition in "symbiotic" relationship they thought to include it.
 - Because this is a cross-cultural study as well the authors should familiarize themselves with the findings that competition and cooperation can go hand in hand especially in collectivistic societies and in cultures characterized by interdependent self-concept (Fülöp, 2004, 2009; Green et al, 2005 etc.).
 - The difference between the type of competition being involved in the experimental and real life competition may also explain the results why the belief in a zero-sum game may not provide additional explanation for competition beyond honor values, because the "honor competition" is not a zero-sum game.
 - On the other hand it is a positive aspect of the research design that competitiveness and cooperation are studied independent of each other so the experimental set up allows to see how they in fact relate to each other, because the authors avoid the false interpretation of not choosing competitive allocations in the first experiment being simply the indicator of cooperativity (what many experimental design applied previously).
 - This is also a limitation of the study: if the experimental, reward-cost mathematical set-up of the study of competitiveness (cooperation as well) in this research can be ecologically valid for non-institutionalized, informal type of competitions involved in honor protective behaviors.
- It is also not sufficiently explained enough why relational mobility was included in the study, what kind of relevance it has in honor behavior and also competition and cooperation.

Sample/Procedure

- The authors write: "Participants were compensated for their participation right after completing the survey and received additional payment based on their own and their paired game partner's decisions at the end of data collection in each society." Please present the exact instruction – what kind of game they play, was contest/cooperation used in the instruction? If in case of the competitive game there was an instruction saying gain as much as you can, or please make moves to win the game etc.
- Participants: educational level, place of residence (urban/rural), age distribution apart from mean age should be presented. These all may have relevance in determining the results and the significantly different distribution among the samples may create differences that are not related to culture but to other demographic indicators. In this respect gender differences should also be presented.

Discussion

- The discussion part is rather descriptive and does not make a sufficient effort to integrate the results and provide a comprehensive "theory" that make sense of the results
- There is this results: "The results showed that among all these societal-level indicators, only GNI ($\beta = -.060$, $p = .047$), and market competitiveness ($\beta = -.063$, $p = .037$) were negatively associated with competition (see Table S14). These results indicated that greater interpersonal competition occurred in harsher environments with lower economic development and productivity." But this result is not integrated with and interpreted together with the following other result: "The results showed that among all these societal-level indicators, only market competitiveness was negatively associated with cooperation ($\beta = -.077$, $p = .022$, see Table S23). One potential explanation could be that increased market competitiveness may encourage

a focus on individual gains over collective benefits, reducing interpersonal cooperation. “ Why does market competition relate negatively to both cooperation and competition? The provided explanations to each of them separately are hard to harmonize and the authors do not make an effort to do so, therefore their results remain fragmented. Their explanation is based on the dichotomous view of cooperation and competition, while it is clear from their results that they are not dichotomous, but there is a need to explain how they can move together.

- It is a very valuable result that the authors were able to decompose the between-society effects into contextual and individual-level effects. They found both in case of competition and cooperation that individuals with the same level of perceived normative and personally endorsed honor values, but inhabiting in societies with higher societal-level honor did not differ, no contextual effects were found. Unfortunately, no explanation was provided for this result either.

- It is also a very interesting and important result that “Personal values of self-promotion and retaliation did not predict cooperation ($p = .342$, see Table S15), but their relationship was negative among societies with lower societal-level honor, which became weaker or even positive as societal-level honor increased (see Table S19 and Figure S3 for simple slope analyses).” But there is no explanation why this is the case, however including in the study 12 different societies that differ along a number of cultural dimensions would help. Maybe it would have been useful to study different cultural dimensions (individualism- collectivism, self-construals, power distance etc.) apart from such macro-level indicators as governmental effectiveness. Many of the studied macro level indicators did not have any connection with competition and cooperations studied in experimental conditions.

- Also very interesting results is the following: “personal values of particularly defense of family reputation becomes more salient in predicting competition in societies where honor norms are less prevalent”. But what kind of explanation the authors have for this?

- These results should be discussed together: On one hand: “The cultural logic of honor tends to emerge in harsh, competitive environments characterized by high status inequality and mobility, and historically weak institutions^{21–23}.” On the other hand: “ Societies characterized by higher mean perceived normative honor values showed higher levels of cooperation ($\beta = .078$, $p = .003$, see Table S15 and Figure 2b) and expectations of interpersonal cooperation ($\beta = .066$, $p = .013$, see Table S16). “ So what is the authors explanation apart from the “nasty neighbor effect”, how members of these societies combine these too?

Overall impression

- All in all, this is a very sophisticated study, with very interesting results but its theoretical and discussion parts need thorough revision.

Reviewer #2 (Remarks to the Author):

In this article, the authors develop a compelling hypothesis about a positive relationship between honor culture and competition, and also explores a relationship between honor culture and cooperation. In particular, the incentivized games used in the research suggests that the research is fundamentally about the relationship of honor culture with competition and cooperation in dyadic social interaction, and especially more about coordination than cooperation given that they use a step-level public goods game between two players as the authors note. The findings are generally clear, and they also yielded some new insights into honor culture, i.e., intriguing differences between two facets of honor culture – self-promotion and retaliation on one hand and defense of family reputation on the other. Given the importance of honor culture in the context of conflict, it is a welcome contribution to the literature. I believe the article is a worthy candidate for *Nature Human Behaviour*; however, there are several aspects of the current manuscript, which I would encourage further consideration in a revision.

First, the introduction section should clarify some of the theoretical concepts more. I think the construct of honor culture is highly complex, and the authors’ description of it does not seem to capture some of the important nuances perhaps necessary to interpret their findings in the later discussion section. For example, how should both personal and family honours should be distinguished in the way they characterized them, and why should they focus on family reputation instead of individual reputation? From memory, Leung and Cohen’s original discussion did not emphasize family reputation as an important facet of honor culture. Another theoretical concept, which I hope the authors can clarify for readers, is the conceptual distinction between the individual-level and societal-level measures of honor and other constructs. The authors’ labels for this distinction is not consistent throughout the article (e.g., normative, intersubjective, etc.) although their use in their data analyses is admirably clear. By discussing the distinction more explicitly and theoretically, the authors may be able to make some of the findings more accessible to readers.

Second, I feel the authors are trying to achieve a little too much in this article. The sections on honor and competition (pp. 10-12) and honor and cooperation (pp. 12-14) report what seems to me to be the focal findings of this research. The results are clear and yield some intriguing new findings about the difference between the two facets of honor culture. However, the section on honor and anticipation of coordination success (pp. 14-15) and honor and behavioral deviations from expectations (pp. 16-17) seem to me to go a little beyond what may be critical for the present article. Don’t get me wrong. These are interesting analyses and the findings are interesting; however, they are very complicated. These sections seem to me to dilute the main message of this research. My impression is that the article would be more impactful if the authors removed these latter sections. Ultimately, however, it is up to the editor and the authors.

Third, the paragraph on the novel finding about the two facets of honor culture (l. 458-469) seems to me to require further elaboration. It raises several questions. For example, how are these two facets related to each other in each country and

overall? If the correlation between them differs across countries, is it appropriate to compute an overall honor culture score by combining these two facets for the societal-level honor culture and use it as a predictor as in Figure 2? In addition, is it appropriate to include both of them as predictors in a mixed-effects model in the way the authors did? These questions need to be addressed either in the discussion section or elsewhere. If this has been addressed somewhere in the article or supplement, I apologize, but I could not find them anywhere. Also, the authors' interpretation that family honor is a kind of public good is intriguing, but I think it needs to be further elaborated. It is a collective good for the family, but it does not seem to me to be a public good for one's family as well as the larger community that includes both the family and other families. Then, those who are concerned about their family's honor may coordinate with their family members, but they do not have to coordinate with non-family members. I might be overinterpreting their argument, but I would appreciate a little more theoretical discussion here.

I hope the authors can further elaborate on another paragraph in the discussion section (l. 470-483). It was not clear to me what the authors meant by "distinguish the general tendency to assign monetary units (and expect others to do so) from the behaviors (and expectations) measured in these two games." I thought that if a participant had a tendency to pay more MUs, then their levels of competition and cooperation as they were measured by these games would both increase. Is this the concern that the authors are trying to address by examining "anticipation of coordination success" and "behavioral deviations from expectations"? If that is the case, the authors' intent did not come through clearly to me. Further, it is not clear to me how the authors' analyses can address the concern. It may be just me, but I think the authors should clarify their concern further or perhaps consider dropping these analyses as I suggested earlier.

Finally, there are a few specific issues I noticed throughout the article.

p. 4 l. 57-58 It may be just me, but I find this sentence a little difficult to follow. When you said, "the afforded actions and outcomes can vary extensively", what do you mean? What actions and outcomes vary between what?

p. 9 l. 187-191 The distinction between perceived normative and personally endorsed honor values, and the distinction between personal honor and family honor can be more clearly described. These constructs are highly complex and the single sentence used here doesn't really convey the complexity.

After reading up to p. 9, it was unclear which variables were being used for the reporting of the results.

p. 10 l. 207-208 What is LRT? Spell it out.

p. 10 l. 208-210 I think you need to clarify what you mean by societal-level and individual-level in the introduction. For most readers, it wouldn't be that obvious. Also, it is interesting that competition and cooperation are correlated at both levels. One possible interpretation is that they both mean paying more.

p. 10 l. 212-213 Here too what the societal-level means may not be so obvious. It would be useful to clarify the distinction between the societal and individual levels earlier in the introduction for all constructs you measured.

p. 10 l. 221-222 I suppose here you are writing about the individual-level measurements of honor values/norms, but that could be more explicitly stated.

p. 10 l. 222-226 It wasn't clear to me why you examined the cross-level interaction between the societal-level honor and the individual-level honor here.

p. 14 l. 312-314 When you said, "the expected sum contribution reached the second point", but do you mean the expected sum contribution equaled the second point? Or did you scored as efficient coordination if the sum was 16 and above?

p. 20 l. 452-453 There seems to be a typo somewhere – this sentence is hard to follow.

p. 22 l. 516-517 Why did you exclude those who did not identify themselves as male or female? Did they have an option to say non-binary or was it a missing value?

p. 24 l. 560-561 "Perceived normative and personal values" wouldn't be clear to most readers.

p. 24 l. 562-564 Can this be a little more clearly described?

p. 24 Contest game – how many rounds of this game did they play?

p. 29 l. 678 "adjusting for response styles"? How did you adjust for response styles?

All in all, I think the article is a strong candidate for publication in *Nature Human Behavior*. However, I hope my comments are of some use to further improve this interesting work, and the authors can address the above issues in a revision before a final editorial decision is made.

Version 1:

Decision Letter:

Our ref: NATHUMBEHAV-24103936A

13th June 2025

Dear Dr. Jin,

Thank you for submitting your revised manuscript "Honour, Competition and Cooperation across 13 Societies" (NATHUMBEHAV-24103936A). It has now been seen by the original referees and their comments are below. As you can see, the reviewers find that the paper has improved in revision. We will therefore be happy in principle to publish it in Nature Human Behaviour, pending minor revisions to satisfy the referees' final requests and to comply with our editorial and formatting guidelines.

We are now performing detailed checks on your paper and will send you a checklist detailing our editorial and formatting requirements within two weeks. Please do not upload the final materials and make any revisions until you receive this additional information from us.

Sincerely,

[REDACTED]

[REDACTED]

[REDACTED]

Nature Human Behaviour

Reviewer #3 (Remarks to the Author):

The authors have done a thorough and conscientious job of addressing the issues I raised in the previous review. In particular, I now think the implications of the exploratory analyses are clearer than before, and I believe they convey useful information. I am happy to recommend publication of the paper. However, I have two additional editorial suggestions. I leave this to the editor, but I think if addressed, they would further enhance the overall readability of the article.

First, although the implications of the exploratory analyses (pp. 14-15) are now better explained in the discussion, the purpose of these analyses was not clear when they were described for the first time. What is there is so descriptive that it isn't really clear why these analyses were carried out, and the reason is provided only in the discussion section. This makes it hard to understand the results. My suggestion is to add a sentence or so to each analysis, so that the purpose of the exploratory analysis is clear from the start. It would be a very minor adjustment, but would improve the readability of the paper quite a bit.

Second, I would add a table in the discussion section that summarizes the main findings. There are two very similar predictors (personal and perceived normative honour values) with two facets at two different levels of analysis. Competition and cooperation are predicted by these with additional cross-level interaction effects etc. etc. These are not easy to keep track of in mind, and having a table that succinctly displays which effects were significant and which was not would help readers interpret their implications. If space permits, this would enhance the impact of the paper in my view.

All in all, I think the article is a fine contribution to the literature. I am happy to support its publication.

Reviewer #4 (Remarks to the Author):

Thank you very much for allowing me to read this fascinating manuscript. I found the manuscript easy to follow and applaud the authors' collective effort, including co-authors from different countries. I think this research is highly relevant, and the results are convincing. Furthermore, I appreciate all the included robustness checks, the transparency, and the illustration of results. The authors even included re-analyses of existing data sets, which can be found in the supplement. I replaced a former reviewer who was no longer available. Thus, I focused on whether the authors addressed the feedback sufficiently when reading the revised manuscript and the reviewers' comments. I added further remarks that the authors could address if they see fit.

The reviewers asked for a more detailed definition and description of central concepts. I think the current version of this manuscript is thorough enough in that respect, but an example here and there might help understanding. For instance, since

honor is a fundamental concept in their work, the authors could add one or two sentences, including examples of self- and other perceptions of honor. The same is true for honor-related norms or behaviors. Also, an example of a society more strongly characterized by a cultural logic of honor would positively influence readability.

The manuscript was improved by adding more information on the ecological validity of the paradigms they used to address a reviewer's comment. I think it would help a reader unfamiliar with economic games if the authors would elaborate more on the kind of social interaction modelled in the two games they used (e.g., real-life examples).

The authors indicate that their samples were stratified by age and gender. Could they say a few words when describing the samples and/or in the discussion section about the representativeness of their samples concerning other demographics, such as education, so that we get a better idea of the generalizability of the findings? Furthermore, if the samples are not representative of these demographics (the sample is likely disproportionate), can the authors briefly report previous findings on the relationship between these factors and honor (e.g., education)? The authors could also include a statement on whether the effects they found potentially under- or overestimate the true effect.

I was surprised to read that the authors excluded participants who did not identify as male or female. This was probably a pre-registered exclusion criterion, so I do not ask the authors to include them in their analyses now. As I noticed this issue independently of a former reviewer, I suggest the authors provide a rationale for each exclusion criterion in the supplementary materials (they already have one in the response letter). Furthermore, they could analyze potential gender differences (not only by adding gender as a control).

I appreciate the authors adding more information on their scales based on the reviewers' comments. However, some of these scales' purposes were still unclear to me. The authors could briefly explain why they included these scales when they mention them in the method section. For example, it makes sense that the authors included a relational mobility scale. Some countries neither cooperate nor compete, so people might be generally less willing to interact with strangers in these cultures.

The authors indicated that 112 participants failed all four comprehension questions in the main paper. Given the recent debate regarding the comprehensibility of economic paradigms (<https://www.sciencedirect.com/science/article/pii/S0167268125001581>), I think it would be interesting to learn how many participants failed one, two, and three comprehension checks and whether there have been differences between countries in that regard.

Minor issues:

This article includes a section called "results," but several results were mentioned before that section.

Typo p. 10 /ln. 214

EDITOR COMMENTS:

1. Our reviewers raise concerns about the conceptualisation of core concepts. In your revisions, please ensure that you use literature-informed clear conceptualisation of the investigated concepts.

Authors' response: We have revised the Introduction and Discussion sections to clarify the conceptualization of core concepts, including competition and cooperation (see pp. 4 to 5, 8), the distinction between individual and family honour (see pp. 8 to 9), as well as personal and perceived normative honour values (see p. 9). For further details of our responses to the reviewers' specific comments, please see our responses to Reviewer 1 comments #2, #3, #4 and #5, and Reviewer 2 comments #2, #4, #7, #10 and #11.

2. Both reviewers have concerns about the presentation of your findings. While Reviewer 1 finds the presentation to be quite descriptive and lacking a broader framework, Reviewer 2 finds that the main message is diluted because of the many analyses and findings. When revising your manuscript, please ensure that you streamline the presentation of the results and findings and place them in a broader theoretical context. We ask that you do not remove findings from the manuscript, especially not when they were pre-registered.

Authors' response: Following Reviewer 1's comments, we have revised the discussion section (pp. 15 to 20) to better explain each finding from the results, connect them to existing theories, and provide a broader framework that integrates findings for both competition and cooperation outcomes. In response to Reviewer 2's concerns, we have emphasized the main findings from our pre-registered analyses in the results (pp. 10 to 13) and discussion sections (pp. 15 to 17), and we have streamlined our coverage of unregistered exploratory analyses that further support and complement these findings (pp. 14 to 15). All findings, whether registered or exploratory, remain included in the manuscript. For further details, please see our responses to Reviewer 1 comments #13, #14, #15, #16, #17, and #18, and Reviewer 2 comments #3 and #5.

3. Reviewer 1 requests that you provide additional methodological details and provide a rationale for the inclusion of individual variables. Please address these requests in full.

Authors' response: We have included additional methodological details as requested (e.g., pp. 23 to 24). Importantly, all study materials, including the instructions for the economic games, are available on the Open Science Framework for peer review and will be made publicly accessible upon publication. We have also explained the rationale for testing additional individual variables, such as beliefs in a zero-sum game for explaining competition and relational mobility for explaining cooperation beyond honour values (p. 9). For further details, please see our responses to Reviewer 1 comments #10 and #11.

REVIEWER COMMENTS:

Reviewer 1

1. This study has the potential to contribute to the deeper understanding of the psychological components of honor culture. The study is a very complex design mainly focusing on the relationship of competitive and cooperative behavioral choices in experimental conditions with perceived normative and personal honor values. All this is done in 12 countries

providing opportunity to highlight similarities and differences among cultures endorsing different honor norms. The study has a number of novel results that are worth to be made available.

The experimental design and the statistical analysis are very complex and adequately serve the purpose of the research goals. My review focuses primarily on the theoretical part and the discussion part because the introductory part and concluding part in my view should be way more elaborated to highlight the relevance of the study and its potential contribution to deeper understanding of the psychology of honor, its relationship with competition and cooperation and in different cultures.

Authors' response 1: Thank you very much for your overall positive assessment of our manuscript. Below, we outline how we addressed your points related to the theoretical and discussion parts. Thank you for the time you put aside in reviewing our manuscript.

Theoretical framework

Concepts: competition, contest, conflict

2. • It would be useful to differentiate between competition and contest. Competition includes both formal/structured (contests) and informal/spontaneous (social comparison based) competitions (Sommer, 1995). Contests are typically institutionalized/formal competitions (e.g. sports, tournaments, beauty contests, presidetial election, applying for management position) in which there are clear criteria and several contestants and limited resources/prizes. Please use the right expressions. This is important because the competition that is involved in honor protecting behavior is not an institutionalized contest, does not have formal rules, clear criteria who is the winner and loser and clear reward system but it is an informal competitive behavior involving many potentially different self-chosen ways to achieve the restoration of the honor of the person or the family. Maybe the word "contest" is valid for the experimental situation involved in this study, but competition is the right word for the interpersonal dynamic that takes place in honor protection. The experimental design is a kind of contest, but honor protection related competition is an informal competition and is not a zero-sum game. If one's honor is protected that does not mean that the other party will have less, or in other words one's gain is not one's cost necessarily.

Authors' response 2: We agree that "competition" is a broader term that encompass both formal and informal type of competitions. In the current study, we indeed used a contest game to model situations that involve a zero-sum conflict of interest, and operationalized competitive behaviours as investing monetary units into the challenge pool as an attempt to take monetary units from their opponent within the game. We revised the manuscript to consistently use the term "competition" when proposing hypotheses about the potential role of the cultural logic of honour in competition, and when interpreting the results and discussing the findings. We used the term "contest" only in the name of the game and acknowledged contest games as one way to study the formal types of competition:

"Contest games are formally structured conflict situations in which one can only be better off at the cost of the other, and one risks being exploited if losing to one's opponent^{43,44}. These games have been used to study formal types of competition, where the incentive structure is defined, and the conflict tends to result in zero-sum outcomes." (p. 6)

The present study only tested competition in a zero-sum situation where one's honour may come from dishonouring the other party, as might occur when male warriors fight for honour to gain status within their tribe (Landes, 2023)¹. However, we also recognize that there could be situations where successful honour protection or restoration does not necessarily result in a loss of honour for another person or party. We have now discussed this in the Discussion section and call for future research to investigate other forms of competitions, including spontaneous competitive behaviours that are not institutionalized by formal zero-sum contests, and to explore how honour may shape these interactions:

“While past research has shown the ecological validity of behaviours measured in economic games⁷⁰⁻⁷³, these insights may not generalize to all social settings⁷⁴. In everyday life, competition (and cooperation) involved in honour-claiming or protecting behaviours may not adhere to formal rules or have an explicit incentive structure to determine winners and losers (provision points of public goods)⁷⁵. Real-life cases of competition may sometimes result in mutual development rather than zero-sum outcomes⁹. Future research could employ methods such as experience sampling to explore the role of honour in shaping spontaneous competition and cooperation in daily social interactions. A further potential methodological limitation is that both competition and cooperation were measured as proactively deciding to invest resources. This approach may introduce confounds to the covariation of competition and cooperation with honour due to a general tendency among individuals to invest monetary units (MUs) into the (challenge/common) pool. On the other hand, this controlled for the potential framing effects that could arise if cooperation were operationalized as “give-some” behaviour (i.e., investing resources) and competition as “keep-some” behaviour (i.e., refraining from investing)⁷⁶.” (pp. 19 to 20)

3. • *Also in the text the authors use competition and conflict interchangeably. “The present study used incentivized economic games to differentiate social interactions in conflict and cooperation situation”. This indicates that they endorse the a concept that competition is always a conflict (zero-sum game) which is not the case, take for example self-developmental competitiveness (Ryckman et al, 1996; Orosz et al,2018) and mutual development competitiveness (Fülöp, 2004 , Fülöp, 2009).*

Authors' response 3: We acknowledge that the contest game used in the current study presented a zero-sum conflict of interest, allowing us to investigate only a specific aspect of competition. We have included sentences in the Discussion section to recognize this limitation, and cited the work following your suggestion to take other types of competition into consideration (pp. 19 to 20). Furthermore, we called for future research to investigate spontaneous competition in daily life using alternative methods (please see our reply to your comment #2 for further details).

Competition and cooperation

4. • *The study has an ongoing theoretical ambiguity if competition and cooperation should be handled together, if they are opposites or not and it quite apparent that the authors are much more familiar with the literature on honor than on the literature and state of the art of competition research. First of all, the fact that they study both competition and cooperation is not fully explained because they provide a short theoretical explanation why honor culture and competition can be related, but they cannot provide a convincing theoretical framework why cooperation should be studied also in relation with honor. It seems they think if*

competition is related than cooperation also has to. They seem to endorse to a degree the already overcome concept that competition and cooperation are opposites because, they realize with “surprise” in their study that they are not.

Authors’ response 4: Thank you for raising these questions. We admit that our thinking was initially shaped by the approach that categorizes competition and cooperation as representing two extremes of a singular behavioural spectrum. Within this literature on economic games, competition and cooperation were often studied within the same game, and researchers have interchangeably used the terms “competition” and “non-cooperation” (as noted in Reviewer 1 comment #8 below). We acknowledge that there is a more recent approach that studies these two behaviours as entwined components harmoniously coexisting or even being positively related in conflicting-interest situations, and that the design of our current study—studying competition and cooperation in separate economic games—is more in line with this recent literature. We thus revised the Introduction to introduce both approaches in the literature and separate games have been used to measure competition and cooperation:

“Past literature has taken different perspectives on studying competition and cooperation. Some research categorizes these behaviours as representing two extremes of a singular behavioural spectrum^{4,6}, while others consider them as entwined components harmoniously coexisting or even being positively related in conflicting-interest situations⁷⁻⁹.” (p. 4)

“To study how the cultural logic of honour may shape both competition and cooperation, we employed two separate incentivized economic games that may provide different opportunities for the expression of honour-related values and norms^{36,37}.” (p. 5)

5. • *In the introduction they write for example: “However, recent findings suggest that a lack of cooperation does not equate to the presence of competition” But this is not recent. The fact that competition and cooperation are not dichotomous and polar opposites have been present in the literature since the 1990’s. Even Morton Deutsch (1990), who was mainly the source of competition and cooperation being “dichotomus” wrote that they are rarely found in their pure form in real life, because for example most conflicts are a mixture of competing and cooperating processes, and the course and consequences of the conflict depend strongly on the nature of this competing-cooperating mixture. Since the 1990s there is an abundant literature on this (e.g. Van de Vliert, 1997) so being “surprised” is for the newcomer in this field ∇. The literature should be important to be familiar with also with the literature on for example cooperative competition (e.g. Fülöp & Takács, 2013). This would also help the authors provide an explanation why it is worth to study how honour as a “cultural logic” affects cooperation. There is not real explanation why cooperation was included – what is its potential psychological/behavioural relationship with honor. It seems that the main idea was to study competition and because formerly the literature tended to handle cooperation and competition in “symbiotic” relationship they thought to include it.*

Authors’ response 5: We have removed this sentence (“*However, recent findings suggest that a lack of cooperation does not equate to the presence of competition*”) and added new sentences to introduce the different perspectives in the past literature. And thank you for suggesting these references, we have cited them accordingly (please see our reply to your comment #4 for further details).

We also added text to the Introduction to provide reasons of studying cooperation in the step-level public goods game in relation to honour:

“Investing in coordinating the successful provision of a public good does not necessarily signify weakness. Unlike contest games where one can only benefit by imposing a cost on others, step-level PGGs give individuals the choice between extending benefits to others at a personal cost or refraining from doing so⁴⁸. The latter enables individuals to express their benevolence, generosity, hospitality, and politeness, which may enhance their own honour and that of their close ingroup^{25,29,49,50}. However, the inherent risk of wasting coordinative efforts may place individuals in a “sucker’s situation” if others do not cooperate, potentially suggesting a negative link between honour and cooperation^{51,52}.” (p. 7)

6. • *Because this is a cross-cultural study as well the authors should familiarize themselves with the findings that competition and cooperation can go hand in hand especially in collectivistic societies and in cultures characterized by interdependent self-concept (Fülöp, 2004, 2009; Green et al, 2005 etc.).*

Authors’ response 6: Thank you for suggesting these references. We now revised the Discussion section and cited these references to explain our findings that honour was positively associated with both competition and cooperation:

“Moreover, our findings suggested that competition and cooperation can coexist within the cultural logic of honour. This aligns with previous research that found self-reliance and group-oriented interdependence to coexist in societies where honour is a central cultural value⁶⁵ and to be associated with competition and cooperation^{66,67}. Our findings suggest that the ecologies fostering the cultural logic of honour may also promote the co-emergence of competition and cooperation.” (p. 17)

7. • *The difference between the type of competition being involved in the experimental and real life competition may also explain the results why the belief in a zero-sum game may not provide additional explanation for competition beyond honor values, because the “honor competition” is not a zero-sum game.*

Authors’ response 7: Thank you for suggesting this potential explanation. We agree that honour related competitions should not be viewed as solely happening in zero-sum situations, and that individuals may engage in more informal and spontaneous competitive behaviours to claim honour in daily life (please see our reply to your comment #2 for further details). The possible explanation we propose here for the finding on beliefs in a zero-sum game is that individual-level perceived normative and personal honour values might already account for the variations in competition observed in zero-sum contests, which might otherwise be attributed to zero-sum beliefs. We have revised Section 3.2.5 of the Supplementary Information (SI) to elaborate more on these findings:

“However, interpretations of the societal-level patterns should be approached with caution due to two key factors: Firstly, the relatively small societal-level sample size (i.e., $N_{\text{society}} = 11$) may limit the statistical power and generalizability of our findings. Secondly, results from the multilevel confirmatory factor analysis indicated that zero-sum beliefs may not vary significantly among the societies sampled in this study (see Section 2.2). At the individual level, we observed that both perceived normative and

personal honour values, especially the facets of self-promotion and retaliation, were positively correlated with zero-sum game beliefs (see Figures S4a and S4b). Hence, perceived normative and personal honour values might already account for that part of the variation in competition that could have been attributed to zero-sum beliefs.” (pp. 28 to 29 in the SI)

8. • *On the other hand it is a positive aspect of the research design that competitiveness and cooperation are studied independent of each other so the experimental set up allows to see how they in fact relate to each other, because the authors avoid the false interpretation of not choosing competitive allocations in the first experiment being simply the indicator of cooperativity (what many experimental design applied previously).*

Authors’ response 8: Thank you for your positive assessment on our design. Indeed, several experimental designs and games from previous research do make that indirect link between competition and cooperation. We have now explicitly introduced the different approaches taken in the past literature and clarified that the current study took the approach to study competition and cooperation in two separate games (please see our reply to your comment #4 for further details).

9. • *This is also a limitation of the study: if the experimental, reward-cost mathematical set-up of the study of competitiveness (cooperation as well) in this research can be ecologically valid for non-institutionalized, informal type of competitions involved in honor protective behaviors.*

Authors’ response 9: While it is true that economic games offer clearly defined cost-benefit mathematical (institutionalized) setups, they are also commonly used to investigate non-institutionalized settings. For example, the dictator game (experimental paradigm where a participants can freely allocate resources between themselves and another participant) is used to study charity behaviour but also simple acts of generosity. Public good games are used to model formal collective action (e.g., paying taxes) as well as more informal non-institutionalized behaviours such as group cooperation. And the same applies for competitive games (e.g., Tullock contests, Chowdhury & Sheremeta, 2011)² that are used to model both formal and informal forms of competitive behaviours.

Previous empirical research has provided some support for the ecological validity of decisions and behaviours measured in economic games across several domains. For example, behaviours observed in social dilemma games have been found to correlate with engagement in socially oriented activities during their studies (Heinz & Schumacher, 2017)³, as well as with various informal cooperative behaviours and outcomes, such as productivity at work (Englmaier & Gebhardt, 2016)⁴, and the management of forest commons (Rustagi et al., 2010)⁵. For competition in conflict situations, while empirical evidence is less abundant, a recent cross-cultural experiment found that behaviours observed in an asymmetric contest game correlated at the country level with the willingness to fight for one’s country (Romano et al., 2022)⁶.

However, we acknowledge in the meantime that insights from economic games may not generalize to all social settings. Whether the economic games used in the current study allow us to generalize over informal and spontaneous types of competition and cooperation is an open question for future research. We have now revised the Discussion section to thoroughly discuss the strengths and limitations associated with the use of the games in the current study:

“We used incentivized economic games to capture participants’ actual behaviours (i.e., beyond hypothetical situations and questionnaire self-reports) as well as their incentivized expectations about other’s behaviours. This approach introduces real consequences for individuals if their reported behaviour does not align with true preferences³⁸. By altering the formal rules of the game, structural variations were applied to study specific types of situations¹¹. For instance, the distinct separation between the contest game and the step-level PGG helped avoid ambiguity in operationalizing competitive and cooperative behaviours⁷. As evidenced by findings from reanalysis of previous datasets, step-level PGGs may be more suitable for measuring cooperation, compared to PDs and continuous PGGs^{14,16}, as the strong appeal of non-cooperation to self-interest in the latter two may limit the expression of the cultural logic of honour in the manifestation of cooperation.

While past research has shown the ecological validity of behaviours measured in economic games⁷⁰⁻⁷³, these insights may not generalize to all social settings⁷⁴. In everyday life, competition (and cooperation) involved in honour-claiming or protecting behaviours may not adhere to formal rules or have an explicit incentive structure to determine winners and losers (provision points of public goods)⁷⁵. Real-life cases of competition may sometimes result in mutual development rather than zero-sum outcomes⁹. Future research could employ methods such as experience sampling to explore the role of honour in shaping spontaneous competition and cooperation in daily social interactions. A further potential methodological limitation is that both competition and cooperation were measured as proactively deciding to invest resources. This approach may introduce confounds to the covariation of competition and cooperation with honour due to a general tendency among individuals to invest monetary units (MUs) into the (challenge/common) pool. On the other hand, this controlled for the potential framing effects that could arise if cooperation were operationalized as “give-some” behaviour (i.e., investing resources) and competition as “keep-some” behaviour (i.e., refraining from investing)⁷⁶.” (pp. 19 to 20)

10. • *It is also not sufficiently explained enough why relational mobility was included in the study, what kind of relevance it has in honor behavior and also competition and cooperation.*

Authors’ response 10: We briefly mentioned in the Introduction the reasons for including beliefs in a zero-sum game and relational mobility in the current study. These analyses were also pre-registered, and we have reported the results in the main text. More detailed information about the rationale for testing beliefs in a zero-sum game as an additional explanation for competition, and relational mobility for cooperation, is available in the SI:

[3.2.5 Additional explanation for competition: Beliefs in a zero-sum game]

“We pre-registered to test whether there are other cultural factors additional to honour values that may explain between-individuals level and between-societies level variation in competition. *Beliefs in a zero-sum game* is the belief that “one person’s gain is possible only at the expense of other persons”⁷. With its roots in classic game theory, zero-sum belief captures the generalized beliefs about the nature of social relations involving completely conflicting interests. Previous research suggests that zero-sum belief can lead to competition and conflict, and varies across societies and social economic status^{7,10}. Therefore, we examined whether beliefs in a zero-sum

game could explain additional variation in competition beyond what was explained by honour values.” (p. 28 in the SI)

[3.3.5 Additional explanation for cooperation: Relational mobility]

“We pre-registered to test whether there are other cultural factors additional to honour values that may explain between-individuals level and between-societies level variation in cooperation. *Relational mobility* is a socio-ecological variable that represents how much freedom and opportunity a society affords individuals to choose and dispose of interpersonal relationships based on personal preference⁸. Low relational mobility societies are characterized by closed networks, and low possibility to change interpersonal relationships and groups. High relational mobility societies are characterized by plenty of opportunities to engage in new friendships based on personal preferences and choices. Past research has found higher cooperation in societies characterized by more flexible and fluid social relations, and that people who perceived their environment to have more opportunities to establish new relationships with strangers were generally more cooperative with strangers¹². We therefore examined whether relational mobility could explain additional variation in cooperation beyond what was explained by honour values.” (p. 49 in the SI)

We have now added the section numbers (Sections 3.2.5 and 3.3.5) to direct readers to the specific locations in the SI.

“As pre-registered, we measured additional variables at the individual level, including beliefs in a zero-sum game⁵⁵ and relational mobility⁵⁶, and obtained society means to construct societal-level indicators for these variables. These variables may offer additional explanations for competition and cooperation, respectively, and have been shown to vary cross-culturally (see Methods and Section 3.2.5 and 3.3.5 in the Supplementary Information, SI, for more details).” (p. 9)

Sample/Procedure

11. • *The authors write: “Participants were compensated for their participation right after completing the survey and received additional payment based on their own and their paired game partner’s decisions at the end of data collection in each society.” Please present the exact instruction – what kind of game they play, was contest/cooperation used in the instruction? If in case of the competitive game there was an instruction saying gain as much as you can, or please make moves to win the game etc.*

Authors’ response 11: In response to a comment from Reviewer 2, we have revised this sentence in the Methods section to include more detailed information on how the payments from the game were calculated:

“After data collection was completed, we randomly selected one out of 12 rounds of participants’ decisions from the two economic games for post hoc decision pairing within each society and calculating participants’ payment from the game^{16,79}. The pairing of decisions was implemented based on both the participant’s gender and the partner’s gender information from the randomly selected round. For example, if a female participant’s game partner in the selected round was male, her decision was paired with a male participant whose game partner was female. The game payment consisted of earnings from making the decision and from making an accurate

estimation of their partner's decision in the selected round. Participants received their game payment within two weeks following the conclusion of data collection.” (p. 23)

To answer your questions, the terms “contest/cooperation” were not used in the instructions. Instead, participants were asked to invest monetary units into the challenge/common pool. We specifically did not instruct participants to gain as much as they can. Instructions were kept consistent across both games, with participants being told, “You and Person B will have the opportunity to earn more MUs by investing between 0 and 10 MUs to the Challenge/Common Pool”. The exact instructions for both games were presented in the study materials PDF file named “materials.pdf”, accessible for peer review on the OSF [<https://rb.gy/n96gmd>] and will be made publicly available upon publication at [<https://osf.io/3dscw/>]. Additionally, the structure of the contest game and the step-level public goods game were described in the method section as follows:

“Contest game. We applied a continuous contest game (also referred to as the rent-seeking game)^{43,80} to measure individuals' own competitive behaviour and expectations of others' competition. The contest game involved two players. Each player received an endowment of 10 MUs and decided how many of the 10 MUs they wanted to invest into a challenge pool (investment = x_i , $0 \leq x_i \leq 10$) or keep for themselves.” (p. 23)

“Step-level public goods game. We applied a step-level public goods game (PGG) to measure cooperation and coordination^{5,53}. This step-level PGG involved two players and two provision points. Each player received an endowment of 10 MUs and decided how many of the 10 MUs they wanted to invest into a common pool (investment = x_i , $0 \leq x_i \leq 10$) or keep for themselves.” (p. 24)

12. • Participants: educational level, place of residence (urban/rural), age distribution apart from mean age should be presented. These all may have relevance in determining the results and the significantly different distribution among the samples may create differences that are not related to culture but to other demographic indicators. In this respect gender differences should also be presented.

Authors' response 12: We previously presented additional summary information about the samples from each society including place of residence, and standard deviation of age, in Table S35 of the SI. Following your suggestions, we have now further enriched Table S35 by including parents' education levels and age ranges. Additionally, we now added a note to Table 1 in the main text directing readers to the SI for more summary demographic information:

“See Table S35 for more summary information on the age range, parents' education level, subjective social status, ethnicity, and living environment (e.g., urban, rural) of the sample from each society.” (p. 31)

During the data collection stage, we have implemented quota groups for participant age (i.e., 18-25, 26-35, 36-45, 46-55, 56+) and gender (i.e., male, female), aiming to recruit equal numbers of participants across these ten quota groups. We have documented in the method section that the recruitment process was “*stratified by age and gender*” (p. 21). In the analyses, participant age and gender have consistently been included in the model as control variables. We suppose it is unlikely that the observed societal-level variations in behaviours

and expectations were due to a skewed distribution of age and gender across different societies.

To address concerns that other demographic variables (parents' education level, place of residence, belongingness to the ethnic majority group in the respective society) may be related to the variations we observed, we conducted robustness checks for the main models that used societal-level honour and individual-level honour indicators to predict competition (see Table S5 and S6) and cooperation (see Table S15 and S16). The results largely replicated findings from Models S5b and S6b for competition, and from Models S15b and S16b for cooperation. We now reported these findings in Sections 3.2.1 and 3.3.1 in the SI (please see below). The syntax used for conducting these analyses were presented in the R markdown file named "analysis_for_revision.Rmd", accessible for peer review on the OSF [<https://rb.gy/n96gmd>] and will be made publicly available upon publication at [<https://osf.io/3dscw/>].

[Section 3.2.1 Honor, competition and expectations about other's competition]
 "For testing individual-level honour indicators, we conducted robustness checks by adding three additional demographic variables as controls into Models S5b and S6b: parents' education levels (1-8), belongingness to the ethnic majority group in the respective society (no, yes), and living environment (rural, urban, both). These analyses were conducted in Models named "m_CGB_2_RC" and "m_CGE_2_RC" in the online syntax "analysis_for_revision.Rmd" on OSF. The results largely replicated findings from Models S5b and S6b. Specifically, perceived normative honour values of self-promotion and retaliation, as well as defence of family reputation, were associated with higher levels of competition [$\beta = .036, p = .048$ (SPR), $\beta = .073, p < .001$ (DFR), see results from Model object "m_CGB_2_RC"]. However, only perceived normative values of defence of family reputation were positively related to expectations of others' competition [$\beta = .028, p = .106$ (SPR); $\beta = .066, p = .001$ (DFR), see results from Model object "m_CGE_2_RC"]. Individual-level measures of personal honour values across both facets were not associated with engagement in competitive behaviour ($ps > .150$)."

[Section 3.3.1 Honor, cooperation and expectations about other's cooperation]
 "For testing individual-level honour indicators, we conducted robustness checks by adding three additional demographic variables as controls into Models S15b and S16b: parents' education levels (1-8), belongingness to the ethnic majority group in the respective society (no, yes), and living environment (rural, urban, both). These analyses were conducted in Models named "m_SLB_2_RC" and "m_SLE_2_RC" in the online syntax "analysis_for_revision.Rmd" on OSF. The results largely replicated findings from Models S15b and S16b. Specifically, perceived normative values of self-promotion and retaliation predicted more cooperation ($\beta = .047, p = .007$, see results from Model object "m_SLB_2_RC"), and perceived normative values of both facets of honour predicted greater expectation of other's cooperation ($\beta = .036, p = .037$ (SPR); $\beta = .068, p < .001$ (DFR), see results from Model object "m_SLE_2_RC"). The two facets of personal honour values showed contrasting associations. Personal values of defence of family reputation positively predicted both cooperation ($\beta = .042, p = .023$) and expectations of others' cooperation ($\beta = .060, p = .002$), while personal values of self-promotion and retaliation negatively predicted expectations of others' cooperation ($\beta = -.043, p = .009$)."

Discussion

13. • *The discussion part is rather descriptive and does not make a sufficient effort to integrate the results and provide a comprehensive “theory” that make sense of the results*

Authors’ response 13: Thank you for raising this question. We have thoroughly revised and restructured the Discussion section to provide a clearer integration of our findings with existing theories and empirical evidence. We now begin by highlighting the main findings regarding the positive association between honour and both competition and cooperation, which support and also further enrich the theoretical framework of the cultural logic of honour (pp. 16 to 17). Following your suggestions, we have elaborated on these associations to present a unified view of competition and cooperation as non-mutually exclusive behaviours, and propose that the ecologies fostering the cultural logic of honour may also promote the co-emergence of competition and cooperation (p. 17). Additionally, our analysis of honour as a multi-faceted and multi-layered cultural construct offers valuable insights for future research, such as the importance of studying honour as a set of normative values that individuals perceive and react to, and examining the cultural context’s affordances when testing the influence of an individual’s personal beliefs or values on their behaviour (pp. 17 to 18). We hope the current version of the discussion has been improved in terms of providing a comprehensive picture interpreting the findings.

14. • *There is this results: “The results showed that among all these societal-level indicators, only GNI ($\beta = -.060$, $p = .047$), and market competitiveness ($\beta = -.063$, $p = .037$) were negatively associated with competition (see Table S14). These results indicated that greater interpersonal competition occurred in harsher environments with lower economic development and productivity.” But this result is not integrated with and interpreted together with the following other result: “The results showed that among all these societal-level indicators, only market competitiveness was negatively associated with cooperation ($\beta = -.077$, $p = .022$, see Table S23). One potential explanation could be that increased market competitiveness may encourage a focus on individual gains over collective benefits, reducing interpersonal cooperation. “ Why does market competition relate negatively to both cooperation and competition? The provided explanations to each of them separately are hard to harmonize and the authors do not make an effort to do so, therefore their results remain fragmented. Their explanation is based on the dichotomic view of cooperation and competition, while it is clear from their results that they are not dichotomous, but there is a need to explain how they can move together.*

Authors’ response 14: We acknowledge that the previous explanations for the market competitiveness were provided independently for competition and cooperation in the SI. We now updated the intercorrelation analyses between societal-level indicators in Table S4a and S4b in the SI, and found that market competitiveness was the strongest societal-level indicator that negatively correlated with societal-level honour (using observed score for societal-level honour: $r = -.971$, see Figure S4a; using factor score for societal-level honour: $r = -.769$, see Figure S4b):

“Based on both observed scores and factor scores of societal-level honour, we report associations between this variable and other societal-level indicators that were retrieved from online databases (see Table S13 for the operationalization of these indicators). We observed substantial negative correlations between societal-level honour and market competitiveness (observed score: $r = -.971$, see Figure S4a; factor

score: $r = -.769$, see Figure S4b), as well as with GNI (observed score: $r = -.909$, see Figure S4a; factor score: $r = -.790$, see Figure S4b), GDP per capita (observed score: $r = -.864$, see Figure S4a; factor score: $r = -.764$, see Figure S4b), and corruption perception index (observed score: $r = -.947$, see Figure S4a; factor score: $r = -.668$, see Figure S4b). Conversely, a large positive correlation was found between societal-level honour and historical prevalence of infectious disease (observed score: $r = .668$, see Figure S4a; factor score: $r = .637$, see Figure S4b).” (p. 78 in the SI)

Based on these findings, we provided new explanations for the negative association between market competitiveness and competition in Section 3.2.6 in the SI, and between market competitiveness and cooperation in Section 3.3.6 in the SI:

“Interestingly, intercorrelations revealed that GNI and market competitiveness were among the strongest societal-level indicators that negatively correlated with societal-level honour (see Figure S4a and S4b). Here, GNI captures the economic wealth of a country. The Global Competitiveness Index (GCI), used to operationalize market competitiveness, is a complex indicator that assesses the ability of countries to provide high levels of prosperity to their citizens. Thus, within the sample of societies included in our dataset, those societies where honour values were perceived to be more prevalent tended to be harsher environments characterized by lower economic development and prosperity, where more interpersonal competition has been observed (see Section 3.3.6 for analyses on the association between other societal-level indicators and cooperation).” (p. 32 in the SI)

“Interestingly, intercorrelations showed that market competitiveness was the strongest societal-level indicator that negatively correlated with societal-level honour, compared to other indicators (see Figure S4a and S4b). The Global Competitiveness Index (GCI) that has been used to operationalize market competitiveness is a highly complex indicator assessing the ability of countries to provide high levels of prosperity to their citizens. Thus, these findings seem to suggest that greater interpersonal cooperation occurred in harsher environments with lower economic productivity and prosperity, where a stronger culture of honour may exist. Considering the results from both competition and cooperation, our findings may imply that harsher environments may require individuals to develop both the ability to cooperate and compete for scarce resources¹³ (see Section 3.2.6 for analyses on the association between other societal-level indicators and competition).” (p. 53 in the SI)

15. • It is a very valuable result that the authors were able to decompose the between-society effects into contextual and individual-level effects. They found both in case of competition and cooperation that individuals with the same level of perceived normative and personally endorsed honor values, but inhabiting in societies with higher societal-level honor did not differ, no contextual effects were found. Unfortunately, no explanation was provided for this result either.

Authors’ response 15: In the previous version of the manuscript, we only briefly explained these findings in the Discussion section. We now revised the same sentence to elaborate more on these findings of the contextual effect:

“We further decomposed the societal-level effects into contextual and individual-level effects, but found no evidence for contextual effects. This suggests that cultural

contexts characterized by varying levels of honour value prevalence may shape interpersonal competition and cooperation primarily through individuals' perceptions of the prescribed values and norms within these contexts." (p. 18)

"Taken together, these findings highlight the importance of examining the cultural logic of honour as a set of normative values that individuals inhabiting different cultural contexts perceive and respond to, ..." (p. 18)

16. • *It is also a very interesting and important result that "Personal values of self-promotion and retaliation did not predict cooperation ($p = .342$, see Table S15), but their relationship was negative among societies with lower societal-level honor, which became weaker or even positive as societal-level honor increased (see Table S19 and Figure S3 for simple slope analyses)." But there is no explanation why this is the case, however including in the study 12 different societies that differ along a number of cultural dimensions would help. Maybe it would have been useful to study different cultural dimensions (individualism- collectivism, self-construals, power distance etc.) apart from such macro-level indicators as governmental effectiveness. Many of the studied macro level indicators did not have any connection with competition and cooperations studied in experimental conditions.*

17. • *Also very interesting results is the following: "personal values of particularly defense of family reputation becomes more salient in predicting competition in societies where honor norms are less prevalent". But what kind of explanation the authors have for this?*

Authors' response 16: Thank you for raising these questions. We address both comments #16 and #17 together here, as they concern findings from the cross-level interactions between personal honour values and societal-level honour. In the previous version of the SI, we provided separate explanations for findings related to competition and cooperation (i.e., Section 3.2.3 for competition, Section 3.3.3 for cooperation). We have now integrated these findings to propose that they may support the concept of cultural affordance, where weaker societal pressure to adhere to honour norms may amplify the influence of personal honour values on behaviour. We did not include analyses on additional cultural dimensions because our prior search did not yield any cultural indicators with 12 observations at the societal level. This raises concerns about a higher risk of model overfitting, should these cultural indicators be included compared to other societal-level indicators in the current study. Furthermore, by combining findings from both competition and cooperation, we believe cultural affordance offers a cohesive explanation for the observed cross-level interactions. We have also added sentences in the main text's discussion section to better explain these results:

"Additionally, findings from cross-level interactions showed that personal honour values were more predictive of competition and cooperation in societies with lower societal-level honour. This suggests that weaker societal pressure to adhere to honour norms may amplify the role of personal honour values in shaping behaviours." (p. 18)

"Taken together, these findings highlight the importance of ..., and considering the affordances cultural contexts provide when testing the role of individual's personal beliefs or values in predicting their behaviours^{40,69}." (p. 18)

"Divergent mechanisms also emerged for the two facets of honour when examining the cross-level interactions in predicting cooperation. In societies with lower (vs. higher) societal-level honour, personally endorsing self-promotion and retaliation was

found to hinder cooperation, while personally endorsing defence of family reputation played a positive role in fostering cooperation.” (pp. 18 to 19)

18. • *These results should be discussed together: On one hand: “The cultural logic of honor tends to emerge in harsh, competitive environments characterized by high status inequality and mobility, and historically weak institutions^{21–23}.” On the other hand: “ Societies characterized by higher mean perceived normative honor values showed higher levels of cooperation ($\beta = .078, p = .003$, see Table S15 and Figure 2b) and expectations of interpersonal cooperation ($\beta = .066, p = .013$, see Table S16). “ So what is the authors explanation apart from the “nasty neighbor effect”, how members of these societies combine these too?*

Authors’ response 17: Our findings suggest that competition and cooperation can coexist within the cultural logic of honour. We acknowledge that in the previous version of the Discussion, we did not provide extensive explanations for these seemingly paradoxical yet reasonable findings. We have now added a paragraph in the Discussion section to elaborate on this:

“We observed a positive association between competition and cooperation at both the societal and individual levels, which supports the perspective that these two processes are not mutually exclusive but coexist^{7,8}. Research increasingly found competition and cooperation to co-occur for the same individuals in group activities⁶², and across domains such as business⁶³ and politics⁶⁴. Similarly, recent evolutionary models that investigated competition and cooperation as independent components have demonstrated the joint evolution of these behaviours⁴⁸. Moreover, our findings suggested that competition and cooperation can coexist within the cultural logic of honour. This aligns with previous research that found self-reliance and group-oriented interdependence to coexist in societies where honour is a central cultural value⁶⁵ and to be associated with competition and cooperation^{66,67}. Our findings suggest that the ecologies fostering the cultural logic of honour may also promote the co-emergence of competition and cooperation.” (p. 17)

Overall impression

19. • *All in all, this is a very sophisticated study, with very interesting results but its theoretical and discussion parts need thorough revision.*

Authors’ response 18: Thank you for recognizing the sophistication and interest of our study. We appreciate your feedback on the theoretical framework and discussion sections. We have undertaken revisions of these sections to better articulate the implications and impact of our findings. We hope these changes address your concerns and enhance the clarity and depth of our manuscript.

Reviewer 2

1. *In this article, the authors develop a compelling hypothesis about a positive relationship between honor culture and competition, and also explores a relationship between honor culture and cooperation. In particular, the incentivized games used in the research suggests that the research is fundamentally about the relationship of honor culture with competition and cooperation in dyadic social interaction, and especially more about coordination than cooperation given that they use a step-level public goods game between two players as the*

authors note. The findings are generally clear, and they also yielded some new insights into honor culture, i.e., intriguing differences between two facets of honor culture – self-promotion and retaliation on one hand and defense of family reputation on the other. Given the importance of honor culture in the context of conflict, it is a welcome contribution to the literature. I believe the article is a worthy candidate for Nature Human Behaviour; however, there are several aspects of the current manuscript, which I would encourage further consideration in a revision.

Authors' response 1: We greatly appreciate your overall positive evaluation of our manuscript. Below, we describe the revisions made in response to your comments. We are thankful for the effort and time you dedicated to reviewing our work.

2. First, the introduction section should clarify some of the theoretical concepts more. I think the construct of honor culture is highly complex, and the authors' description of it does not seem to capture some of the important nuances perhaps necessary to interpret their findings in the later discussion section. For example, how should both personal and family honours should be distinguished in the way they characterized them, and why should they focus on family reputation instead of individual reputation? From memory, Leung and Cohen's original discussion did not emphasize family reputation as an important facet of honor culture. Another theoretical concept, which I hope the authors can clarify for readers, is the conceptual distinction between the individual-level and societal-level measures of honor and other constructs. The authors' labels for this distinction is not consistent throughout the article (e.g., normative, intersubjective, etc.) although their use in their data analyses is admirably clear. By discussing the distinction more explicitly and theoretically, the authors may be able to make some of the findings more accessible to readers.

Authors' response 2: Thank you for raising the questions about (a) the necessity of distinguishing between individual honour and family honour, and (b) the conceptual distinction between individual-level and societal-level measures. We acknowledge the complexity of the honour construct, shaped by an individual's self-view and their social reputation. Honour also has a relational nature, where threats to or enhancements of honour can impact close others or affiliated social groups. This often leads individuals to preemptively or retaliatively defend their reputation or that of their close ingroup.

To address your first question, we have revised the Introduction to clarify why it is crucial to consider not only individual honour but also family honour:

“Honour can be understood as the value of a person in their own eyes and in the eyes of others²¹. To be honourable, individuals must actively express certain traits or behaviours to claim honour and gain recognition and respect from others in their social environment²²⁻²⁴. Recently, honour has been studied as a cultural logic comprising shared beliefs, values, norms, and practices that cohere around the central theme of pursuing honour²⁵. This cultural logic tends to emerge in harsh, competitive environments characterized by status inequality and instability, and historically weak institutions²⁶⁻²⁸. In these environments, individuals likely develop strategies to protect their safety and resources, as well as those of their close ingroups such as family members, through personal actions. A reputation for toughness and strength is adaptive because it can deter competitors and prevent being exploited in the future^{25,27,29}. Individuals' willingness to retaliate or even pre-emptively defend themselves, securing a tough reputation, can be selected as an important survival

strategy and thus become normative in these environments³⁰. Moreover, individuals may engage in similar actions to defend the honour of their close others or affiliated social groups (e.g., typically family members)³¹. However, the pursuit of honour seems to risk escalating unnecessary conflict, especially among unrelated individuals. Past literature has documented that honour-related norms and behaviours can foster conflict responses such as violence, aggression and honour-related crimes^{27,32-35}.” (p. 5)

We now also introduced why we were interested in measuring both the individual and family facets of honour in the current study:

“Here, we assessed both individual and family (i.e., close ingroup) facets of honour because these two facets may have different implications for social interactions within the cultural logic of honour. Specifically, our measure of individual honour focused on valuing actions of *self-promotion and retaliation* to claim honour, whereas our measure of family honour mainly focused on the *defence of family reputation*^{42,54}. Compared to the family facet, individual honour may be theoretically more relevant for shaping decisions in the dyadic interactions captured in the current study. However, empirical research into the implications of family honour remains limited so far. We sought to contribute to this literature by testing whether the degree to which individuals value defending the honour shared by their family shapes their interactions with unrelated others in their society.” (p. 8)

To address your second question, we have revised the Introduction to clearly distinguish between individual-level and societal-level measures of honour and other constructs. Regarding terminology consistency, we have used “perceived normative honour values” throughout the manuscript to describe the measured variable. But in the meantime, we retained the term “intersubjective” in a few places in tandem with the term “perceived normative honour values”. This terminology could help link our findings to prior research on “intersubjective culture” by Chiu and colleagues (2010)⁷, who have employed similar measurement methods. We have also ensured that the use of the term “intersubjective” was accompanied by a citation of Chiu’s work to maintain clarity:

“We operationalized the cultural logic of honour through the individual-level measures of personal endorsement of the abovementioned two facets of honour values (referred to as *personal values*) as well as intersubjective perceptions of how prevalent the two facets of honour values are within each society (referred to as *perceived normative values*)^{40,41}. The society mean of perceived normative honour values across both facets was used to construct a societal-level indicator, characterizing the extent to which a society can be considered a culture of honour (referred to as *societal-level honour*). As pre-registered, we measured additional variables at the individual level, including beliefs in a zero-sum game⁵⁵ and relational mobility⁵⁶, and obtained society means to construct societal-level indicators for these variables. These variables may offer additional explanations for competition and cooperation, respectively, and have been shown to vary cross-culturally (see Methods and Section 3.2.5 and 3.3.5 in the Supplementary Information, SI, for more details).” (p. 9)

3. Second, I feel the authors are trying to achieve a little too much in this article. The sections on honor and competition (pp. 10-12) and honor and cooperation (pp. 12-14) report what

seems to me to be the focal findings of this research. The results are clear and yield some intriguing new findings about the difference between the two facets of honor culture. However, the section on honor and anticipation of coordination success (pp. 14-15) and honor and behavioral deviations from expectations (pp. 16-17) seem to me to go a little beyond what may be critical for the present article. Don't get me wrong. These are interesting analyses and the findings are interesting; however, they are very complicated. These sections seem to me to dilute the main message of this research. My impression is that the article would be more impactful if the authors removed these latter sections. Ultimately, however, it is up to the editor and the authors.

Authors' response 3: We acknowledge that the sections on “honour and anticipation of coordination success” and “honour and behavioural deviations from expectations” were not pre-registered and served as exploratory analyses. Nevertheless, we believe that these analyses provide complementary evidence that supported the main findings. For example, we found that the self-promotion and retaliation facet of honour was relevant for efficient competition, while the defence of family reputation facet was relevant for conditional cooperation. Furthermore, these findings provide empirical evidence that complements the theoretical framework of the cultural logic of honour and may also offer directions for future research. For example, these findings suggested that perceptions of self-promotion and retaliation in one's society may encourage a reasonable and efficient level of competition, rather than excessive competition. And while family honour norms may discourage contributing more than others, they may play an important role in advocating fairness and reciprocal principles in cooperation.

Based on your suggestion, while we still retain these exploratory analyses in the Results section of the main text (following the Editor's request not to remove any findings from the manuscript), we have streamlined the presentation and restructured them into a single subsection titled “Exploratory analyses: Honor and behaviours adjusted by expectations”. This change emphasizes their role as supportive explorations, not primary analyses. We have also revised and shortened these subsections to outline the specific questions these exploratory analyses could address:

“(Less-)efficient coordination success. We explored how the cultural logic of honour relates to individuals' anticipation of coordination dynamics and outcomes with unrelated others in their society (see Methods).” (p. 14)

“(Less-)efficient competition. We distinguished different patterns of competitive behaviour based on the anticipated outcomes in the contest game (see Methods), exploring how the cultural logic of honour relates to *efficient competition* (defined as minimizing costs to win the contest) and *less-efficient competition* (defined as overspending to win the contest).” (p. 14)

“(Un)conditional cooperation. We also distinguished different types of cooperative behaviour which may have reflected different underlying motives (see Methods), exploring how the cultural logic of honour relates to *conditional cooperation* (defined as matching the expected contribution of one's partner in the same round) and *unconditional cooperation* (defined as exceeding the expected contribution of one's partner in the same round).” (p. 15)

We have further revised the Discussion section to elaborate on how each of these exploratory analyses added valuable insights that support our main findings:

[(Less-)efficient coordination success]

“The positive association between perceived normative honour values and cooperation—including evidence from levels of cooperation, coordinative decisions targeting achieving efficient coordination (e.g., contributing 8 MUs), and anticipation of coordination success—both at societal and individual levels, aligns with earlier research on honour cultures and conflict management. This research found that individuals from honour, compared to non-honour, cultures were more willing and able to handle conflict situations constructively, and made more cooperative offers in negotiations when the situation afforded such opportunities — such as in the absence of insults⁵⁹, or in the presence of social rewards⁴⁹.” (pp. 16 to 17)

[(Less-)efficient competition]

“Interestingly, exploratory analyses suggested that individuals who perceived stronger normative values of self-promotion and retaliation may aim to avoid excessive spending to win a contest, rather than engage in inefficient competition that would diminish their welfare after winning. This finding questions claims in the literature linking honour with abhorring cost-benefit calculations²⁵. When competition is institutionalized with clearly defined incentive structure, such conditions afford honour-related norms to manifest in efforts to minimize costs to win, based on expectations of the other’s competition.” (p. 16)

[(Un)conditional cooperation]

“Moreover, exploratory analyses that subtracted expectations of others’ cooperation from one’s own suggested that individuals who perceived stronger normative values of defence family reputation may be more likely to condition their own cooperation on the expected cooperation of others, but less likely to respond altruistically to expected less-cooperative others. These findings provided empirical support for the theorised importance of positive reciprocal principles and self-protection to avoid being exploited in social interactions within the cultural logic of honour²⁵.” (p. 17)

4. Third, the paragraph on the novel finding about the two facets of honor culture (l. 458-469) seems to me to require further elaboration. It raises several questions. For example, how are these two facets related to each other in each country and overall? If the correlation between them differs across countries, is it appropriate to compute an overall honor culture score by combining these two facets for the societal-level honor culture and use it as a predictor as in Figure 2? In addition, is it appropriate to include both of them as predictors in a mixed-effects model in the way the authors did? These questions need to be addressed either in the discussion section or elsewhere. If this has been addressed somewhere in the article or supplement, I apologize, but I could not find them anywhere. Also, the authors’ interpretation that family honor is a kind of public good is intriguing, but I think it needs to be further elaborated. It is a collective good for the family, but it does not seem to me to be a public good for one’s family as well as the larger community that includes both the family and other families. Then, those who are concerned about their family’s honor may coordinate with their family members, but they do not have to coordinate with non-family members. I might be overinterpreting their argument, but I would appreciate a little more theoretical discussion here.

Authors' response 4: Thank you for raising these insightful questions. Below, we provide our responses to each question.

1) *Is it appropriate to compute an overall honour culture score by combining the two facets of honour?*

We considered constructing a societal-level indicator for cultures of honour by combining the two facets of honour based on both theoretical and empirical considerations.

From the theoretical point of view, the socio-ecological approach suggests that honour cultures tend to emerge from conditions characterized by economic precariousness, independent subsistence styles, and historically weaker institutions. These conditions shape honour-related values, beliefs, norms, and practices as adaptations to the environment, making them more prevalent in certain contexts. In our manuscript, we argue that pursuing honour for both individuals themselves and for their close ingroups, particularly family members, is integral within the cultural logic of honour:

“Honour can be understood as the value of a person in their own eyes and in the eyes of others²¹. To be honourable, individuals must actively express certain traits or behaviours to claim honour and gain recognition and respect from others in their social environment²²⁻²⁴. Recently, honour has been studied as a cultural logic comprising shared beliefs, values, norms, and practices that cohere around the central theme of pursuing honour²⁵. This cultural logic tends to emerge in harsh, competitive environments characterized by status inequality and instability, and historically weak institutions²⁶⁻²⁸. In these environments, individuals likely develop strategies to protect their safety and resources, as well as those of their close ingroups such as family members, through personal actions. A reputation for toughness and strength is adaptive because it can deter competitors and prevent being exploited in the future^{25,27,29}. Individuals' willingness to retaliate or even pre-emptively defend themselves, securing a tough reputation, can be selected as an important survival strategy and thus become normative in these environments³⁰. Moreover, individuals may engage in similar actions to defend the honour of their close others or affiliated social groups (e.g., typically family members)³¹. However, the pursuit of honour seems to risk escalating unnecessary conflict, especially among unrelated individuals. Past literature has documented that honour-related norms and behaviours can foster conflict responses such as violence, aggression and honour-related crimes^{27,32-35}.” (p. 5)

Empirically, we have developed a societal-level indicator of honour by calculating mean perceived normative honour values across the two facets for each society, and also used multilevel confirmatory factor analysis (CFA) to obtain factor scores for honour at both the between-society and within-society levels as a robustness check. The results of multilevel CFA provided model fitting evidence supporting the presence of one societal-level content factor of honour and two individual-level content factors (i.e., defence of family reputation, self-promotion and retaliation). This measurement model replicates previously published findings using two nonoverlapping multinational datasets collected across a similar range of countries (Vignoles et al., 2024, Studies 1 and 2)⁸. More details can be found in Sections 2.1 of the SI:

“For both personal and perceived normative honour values, we adopted a multilevel measurement model based on recent research⁶. This measurement model included one culture-level content factor of honour, and separated honour values into two distinct content factors, (a) defending family reputation and (b) self-promotion and retaliation, at the individual level...The model fitted the data well (perceived normative values: $\chi^2_{[161]} = 880.721$, $CFI = .958$, $TLI = .952$, $RMSEA = .036$, $SRMR_{Within} = .114$, $SRMR_{Between} = .145$; personal values: $\chi^2_{[161]} = 898.684$, $CFI = .958$, $TLI = .953$, $RMSEA = .037$, $SRMR_{Within} = .082$, $SRMR_{Between} = .173$). The between-society level content factor of honour values showed significant variance in the multilevel CFA model for both personal ($p = 0.016$) and perceived normative values ($p = 0.020$), indicating cross-societal variation in both personal and perceived normative honour values. Factor scores for societal-level honour values and individual-level honour values, the latter including the dimensions of defence of family reputation, and self-promotion and retaliation, were saved from the final CFA models for personal and perceived normative honour values (see Mplus syntax file “personal_honor_values.out” and “perceived_normative_honor_values.out” on OSF at <https://rb.gy/n96gmd>). ” (p. 8 in the SI)

2) *Is it appropriate to include both facets of individual-level honour indicators as predictors in a mixed-effects model?*

In Table S2 (#2) of the SI, we have reported the unregistered decision of including individual-level measures of both facets of honour values as predictors into the same model (detailed in the column “Deviation Description”). The potential advantages of this approach were also briefly discussed in the column “Reader Impact”:

“Deviation Description: For individual-level analyses, we calculated two separate indicators for each facet of perceived normative honour values as well as of personal honour values. These four individual-level indicators for perceived normative and personal honour values were then entered as predictors to the same mixed-effects models for hypotheses testing and additional analyses.

Reader Impact: This unregistered step allowed for a more precise analysis of the relationships between perceived normative (and personal) honour values and the outcome variables. This approach prevented the dilution of specific associations that might occur with an overall score, enabling a clearer understanding of how each facet uniquely explained variations in behaviours and expectations.” (p. 3 in the SI)

To confirm the statistical appropriateness of including all four individual-level honour indicators as predictors in the models predicting competition or cooperation, we have conducted multicollinearity tests. The findings were reported in the Sections 3.2.1 and 3.3.1 of the SI:

“The generalized variance inflation factor adjusted for the degree of freedom indicated a low risk of multicollinearity in both models [all the $GVIF^{1/(2 \times Df)} < 2$] (see Models “m_CGB_2_vif” and “m_CGE_2_vif” in the online syntax “data_analysis.Rmd” on OSF).” (p. 15 in the SI)

“The generalized variance inflation factor adjusted for the degree of freedom indicated a low risk of multicollinearity in both models [all the $GVIF^{1/(2 \times Df)} < 2$] (see

Models “m_SLB_2_vif” and “m_SLE_2_vif” in the online syntax “data_analysis.Rmd” on OSF).” (p. 36 in the SI)

Following your suggestions, we have added more information in the “Analytic Strategy” subsection of the Methods section. This addition clarifies the four individual-level honour indicators and explains why they were included as predictors in the same model:

“We calculated separate indicators of each facet of perceived normative honour values as well as of personal honour values, and simultaneously included all four individual-level honour indicators as predictors into the mixed-effects model. This approach allowed us to test the roles of perceived normative values and personal values while controlling for one another, as well as to examine how each facet uniquely explained variation in behaviours and expectations.” (p. 28)

3) *Is there a more appropriate interpretation for the role of family honour in cooperation?*

We acknowledge that the interdependent and coordinative nature of family honour may not fully explain the positive association we observed between personal values of defence of family reputation and cooperation. However, this remains an intriguing area for future research. Accordingly, we have revised the Discussion section to elaborate on these findings and to encourage further studies on defending honour of those larger ingroups beyond the family:

“One possible explanation lies in the interdependent and coordinative nature of family honour—a family’s honour is maintained by members working together to uphold their family’s reputation and prevent any damage to it in the surrounding environment²⁹. However, it remains unclear why this family honour-oriented coordination motive extended beyond close ingroup boundaries to also benefit unrelated others within the same society (in the absence of any outgroup from other societies). Future research could examine personal values of defending the honour of larger ingroups beyond the family to determine whether the same patterns hold at varying levels of group boundaries.” (p. 19)

5. I hope the authors can further elaborate on another paragraph in the discussion section (l. 470-483). It was not clear to me what the authors meant by “distinguish the general tendency to assign monetary units (and expect others to do so) from the behaviors (and expectations) measured in these two games.” I thought that if a participant had a tendency to pay more MUs, then their levels of competition and cooperation as they were measured by these games would both increase. Is this the concern that the authors are trying to address by examining “anticipation of coordination success” and “behavioral deviations from expectations”? If that is the case, the authors’ intent did not come through clearly to me. Further, it is not clear to me how the authors’ analyses can address the concern. It may be just me, but I think the authors should clarify their concern further or perhaps consider dropping these analyses as I suggested earlier.

Authors’ response 5: Thank you for suggesting a potential explanation for the positive correlation between competition and cooperation. We had speculated similarly but had not previously detailed this in the Discussion. We have now added a paragraph to discuss these findings further and have also acknowledged this methodological consideration as a limitation:

“We observed a positive association between competition and cooperation at both the societal and individual levels, which supports the perspective that these two processes are not mutually exclusive but coexist^{7,8}. Research increasingly found competition and cooperation to co-occur for the same individuals in group activities⁶², and across domains such as business⁶³ and politics⁶⁴. Similarly, recent evolutionary models that investigated competition and cooperation as independent components have demonstrated the joint evolution of these behaviours⁴⁸.” (p. 17)

“A further potential methodological limitation is that both competition and cooperation were measured as proactively deciding to invest resources. This approach may introduce confounds to the covariation of competition and cooperation with honour due to a general tendency among individuals to invest monetary units (MUs) into the (challenge/common) pool. On the other hand, this controlled for the potential framing effects that could arise if cooperation were operationalized as “give-some” behaviour (i.e., investing resources) and competition as “keep-some” behaviour (i.e., refraining from investing)⁷⁶.” (p. 20)

Regarding the exploratory analyses on “behavioural deviations from expectations”, we have revised the relevant subsections to clearly outline the specific questions these analyses could address. For more details on our rationale for including these in the main text, please refer to our response to your comment #3. We note also that the Editor gave us a strong steer not to drop any findings from the manuscript. We hope you find these revisions satisfactory and agree that these exploratory findings intriguingly enhance our understanding of how honour relates to competition and cooperation.

Finally, there are a few specific issues I noticed throughout the article.

6. p. 4 l. 57-58 It may be just me, but I find this sentence a little difficult to follow. When you said, “the afforded actions and outcomes can vary extensively”, what do you mean? What actions and outcomes vary between what?

Authors’ response 6: In this sentence, we aim to communicate that the actions each individual can take and the outcomes they may receive, can vary across social situations involving conflicts of interest. In the following sentences, we provided two examples: actions can range from choosing to compete or not, to deciding whether to coordinate or not, while outcomes can vary from zero-sum to positive-sum scenarios. We have revised this sentence to align with the examples that follow:

“Social interactions frequently involve conflicts of interest between individuals, where the actions available to individuals (e.g., competition, cooperation) and the outcomes they might receive (e.g., zero-sum, positive-sum) can vary extensively¹⁻³.” (p. 4)

7. p. 9 l. 187-191 The distinction between perceived normative and personally endorsed honor values, and the distinction between personal honor and family honor can be more clearly described. These constructs are highly complex and the single sentence used here doesn’t really convey the complexity.

Authors’ response 7: We agree that these constructs are highly complex and have revised this paragraph for greater clarity. To avoid confusion associated with the term “personal”, we

now distinguish between “individual honour” and “family honour” for the two facets of honour (pp. 8 to 9). Additionally, we use “personal honour values” and “perceived normative honour values” to differentiate between measures of personal endorsement and intersubjective perceptions (p. 9). Please also refer to our reply to your comment #2 for further details.

8. After reading up to p. 9, it was unclear which variables were being used for the reporting of the results.

Authors’ response 8: Related to our reply to your comment #7, we have revised the Introduction to clarify that: (a) at the individual-level, we measured two facets of personal honour values (self-promotion and retaliation, defence of family reputation), and the same two facets of perceived normative honour values, and (b) at the societal-level, we generated an overall societal-level honour indicator using perceived normative honour values across the two facets. Therefore, there were four individual-level honour variables and one societal-level honour variable used in the analysis and reporting of the results.

9. p. 10 l. 207-208 What is LRT? Spell it out.

Authors’ response 9: By LRT, we were referring to the Likelihood Ratio Test which has been conducted to test whether including the between-society variance as a random intercept significantly improved the model fit compared to a simpler model without this parameter. To clarify the statistical method used, we have replaced “LRT” with the chi-square (χ^2) symbol to represent the distribution of the test statistic for a Likelihood Ratio Test (LRT):

“We observed significant differences across societies in competition and cooperation, with between-society variance significantly different from zero for competition, $\chi^2(1) = 31.30, p < .001$, and cooperation, $\chi^2(1) = 39.80, p < .001$ (see Table S3).” (p. 10)

10. p. 10 l. 208-210 I think you need to clarify what you mean by societal-level and individual-level in the introduction. For most readers, it wouldn’t be that obvious. Also, it is interesting that competition and cooperation are correlated at both levels. One possible interpretation is that they both mean paying more.

11. p. 10 l. 212-213 Here too what the societal-level means may not be so obvious. It would be useful to clarify the distinction between the societal and individual levels earlier in the introduction for all constructs you measured.

Authors’ response 10: Thank you for suggesting that we clarify the terms “societal-level” and “individual-level” in the introduction. We have differentiated between these two levels both in the paragraph before proposing our hypotheses and in the paragraph describing how honour was operationalized:

“To study how the cultural logic of honour may shape both competition and cooperation, we employed two separate incentivized economic games that may provide different opportunities for the expression of honour-related values and norms^{36,37}. Economic games are highly structured situations with formal rules and unambiguous outcomes, which are nonetheless widely used to study human judgement, decision-making and behavioural choices that may transfer into everyday life^{36,38}. We examined how individuals’ behaviour in these games may be predicted

by honour values on multiple levels: societal-level variation in honour culture (i.e., effects of living in societies where honour values are more or less prevalent)³⁹, individual-level variation in perceived societal honour norms (i.e., effects of perceiving honour values as more or less normative in one's society—also known as “intersubjective culture”)⁴⁰⁻⁴², and individual-level variation in personal honour values (i.e., effects of personally internalizing cultural values of honour more or less)²⁵.” (pp. 5 to 6)

“We operationalized the cultural logic of honour through the individual-level measures of personal endorsement of the abovementioned two facets of honour values (referred to as *personal values*) as well as intersubjective perceptions of how prevalent the two facets of honour values are within each society (referred to as *perceived normative values*)^{40,41}. The society mean of perceived normative honour values across both facets was used to construct a societal-level indicator, characterizing the extent to which a society can be considered a culture of honour (referred to as *societal-level honour*). As pre-registered, we measured additional variables at the individual level, including beliefs in a zero-sum game⁵⁵ and relational mobility⁵⁶, and obtained society means to construct societal-level indicators for these variables. These variables may offer additional explanations for competition and cooperation, respectively, and have been shown to vary cross-culturally (see Methods and Section 3.2.5 and 3.3.5 in the Supplementary Information, SI, for more details).” (p. 9)

Regarding the interesting findings on the positive correlation between competition and cooperation, indeed, it could be due to that both competition and cooperation were measured as investing more, albeit for different purposes. In our reply to your comment #5, we have added a paragraph to discuss these findings further and have also acknowledged this methodological consideration as a limitation.

12. p. 10 l. 221-222 I suppose here you are writing about the individual-level measurements of honor values/norms, but that could be more explicitly stated.

Authors' response 11: Yes, we have now revised this sentence to explicitly state that it is individual-level analyses:

“Individual-level measures of personal honour values across both facets were not associated with engagement in competitive behaviour (*H3: ps* > .146, see Table S5).” (p. 10)

13. p. 10 l. 222-226 It wasn't clear to me why you examined the cross-level interaction between the societal-level honor and the individual-level honor here.

Authors' response 12: We added a sentence in the Results section to explain the reason for conducting the cross-level interaction:

“We also explored the potential interaction between individual-level personal honour values and societal-level honour, as the as the implications of personally endorsing honour values could differ according to the broader cultural logic in one's society.” (p. 10)

14. p. 14 l. 312-314 *When you said, “the expected sum contribution reached the second point”, but do you mean the expected sum contribution equaled the second point? Or did you score as efficient coordination if the sum was 16 and above?*

Authors’ response 13: We scored efficient coordination if the expected sum contribution was 16 MUs or more, and less-efficient coordination if it was 12 MUs or more but fewer than 16 MUs. We have revised this sentence to better explain the categorization of a game round:

“In the step-level PGG, we also distinguished different types of anticipation of successful coordination by summing up an individual’s cooperation and their expectations of their game partner’s cooperation. Specifically, we categorized a given round as *efficient coordination* if the expected sum contribution reached the second provision point (i.e., 16 MUs or more), as *less-efficient coordination* if it only reaches the first provision point (i.e., 12 MUs or more but fewer than 16 MUs), and otherwise as *failed coordination* (i.e., fewer than 12 MUs, see Figure 3).” (pp. 25 to 26)

15. p. 20 l. 452-453 *There seems to be a typo somewhere – this sentence is hard to follow.*

Authors’ response 14: Thank you for pointing this out. Indeed, the content “Beyond honor values, the current research also identified” was included erroneously. We have now removed this content from the sentence.

16. p. 22 l. 516-517 *Why did you exclude those who did not identify themselves as male or female? Did they have an option to say non-binary or was it a missing value?*

Authors’ response 15: Yes, participants were given a third option when asked to self-report their sex/gender at the beginning of the study, following the provision of informed consent. The options were “male”, “female”, and “not listed, please specify”. During the translation of the survey into Arabic (Egypt, Lebanon, Morocco), Greek (Greece, Greek Cypriot community), Japanese (Japan), Korean (South Korea), and Turkish (Türkiye, Turkish Cypriot community), the terms “sex” and “gender” were translated using the same word. In the English (U.K., U.S.A.), Italian (Italy), and Spanish (Spain) versions, the term “sex” was used. Despite the various translations, across these 13 societies, we consider this self-reported measure to align more closely with a working definition of gender rather than a strictly binary concept of sex, as defined by the Nature portfolio. This is because participants were provided with a third option to express identities beyond male and female.

Throughout the data collection stage, we were inclusive in inviting participants to take part in the study. All participants, regardless of the sex/gender option selected, were allowed to complete the survey and were compensated for their participation and any earnings from the game. However, in the data analysis stage, we only included participants who self-identified as male or female. This decision was made to align analysis with our experimental design, which manipulated the gender information of the game partner based on binary gender categories (male, female, gender information not provided) to explore specific dynamics in competition or cooperation behaviours and expectations among female-female, male-male, and mixed-gender pairs. The “gender information not provided” category allowed us to explore interactions when a partner’s gender was unspecified, serving as a reference for behaviours and expectations towards strangers in general. Given the scope of this manuscript, the detailed exploration of how participant gender and partner gender relate to competition

and cooperation will be addressed in a separate paper. For the purposes of this study, participant gender and partner gender were included as control variables in our models.

17. p. 24 l. 560-561 “*Perceived normative and personal values*” wouldn’t be clear to most readers.

Authors’ response 16: We have revised this sentence to differentiate between “perceived normative values” and “personal values”:

“Participants also completed several measures, including perceived normative values and personal values across the two facets of honour (i.e., self-promotion and retaliation, defence of family reputation), beliefs in a zero-sum game, and relational mobility. They were debriefed at the end of the experiment and compensated for their participation through the panel provider/research agency.” (p. 23)

18. p. 24 l. 562-564 *Can this be a little more clearly described?*

Authors’ response 17: We have now provided a more detailed description of how the game payments were calculated for each participant:

“After data collection was completed, we randomly selected one out of 12 rounds of participants’ decisions from the two economic games for post hoc decision pairing within each society and calculating participants’ payment from the game^{16,79}. The pairing of decisions was implemented based on both the participant’s gender and the partner’s gender information from the randomly selected round. For example, if a female participant’s game partner in the selected round was male, her decision was paired with a male participant whose game partner was female. The game payment consisted of earnings from making the decision and from making an accurate estimation of their partner’s decision in the selected round. Participants received their game payment within two weeks following the conclusion of data collection.” (pp. 23)

19. p. 24 *Contest game – how many rounds of this game did they play?*

Authors’ response 18: The number of rounds for each game is reported in the “Procedure and experimental design” subsection of the Methods section:

“Participants were asked to make six independent rounds of decisions in the contest game, and another six rounds in the step-level public goods game. Each round involved a different game partner—either male, female, or with gender information not provided—from their own society, whose decisions were asynchronously paired with those of the participant after the experiment.” (p. 22)

20. p. 29 l. 678 “*adjusting for response styles*”? *How did you adjust for response styles?*

Authors’ response 19: This information was detailed in Sections 2.1 to 2.3 of the SI:

[2.1 Perceived normative and personal honour values]

“To adjust for differences in response style in the measurement model, we created four indicators of acquiescent responding, each defined by averaging a pair of items

with opposing substantive content from the relational mobility measure (e.g., averaged agreement with “*It is common for these people to have a conversation with someone they have never met before*” and “*It is uncommon for these people to have a conversation with people they have never met before*” without reverse scoring). We used these items to anchor a method factor which allowed us to adjust our measures of perceived normative values (and personal values) for acquiescent responding both at the individual and cultural level.” (p. 8 in the SI)

[2.2 Belief in a zero-sum game]

“The same approach was used to adjust for differences in response style as was used for honour values, i.e., four indicators of acquiescent responding were created from selected items of the relational mobility measure to anchor method factors at the within- and between-samples level (see 2.1 Personal and perceived normative honour values). Two items (“*When some people are getting poorer, it means that other people are getting richer*”, “*The wealth of a few is acquired at the expense of many*”) were retained only at the within, but not the between-society level due to negative loadings and these items were therefore centred within societies.” (p. 9 in the SI)

[2.3 Relational mobility]

“To adjust for differences in response style, we introduced a method factor at both within and between-samples levels to account for variance due to acquiescent responding; all items had a fixed loading of 1 on this factor.” (p. 10 in the SI)

In the Methods section, we have pointed readers to these specific sections for the relevant information:

“To ensure the robustness of our analyses, we also obtained factor scores for honour values at both the between-society and within-society levels using confirmatory factor analysis adjusting for response styles in Mplus 8.10⁸⁵ (see Section 2 in the SI for more information).” (p. 29)

21. All in all, I think the article is a strong candidate for publication in Nature Human Behavior. However, I hope my comments are of some use to further improve this interesting work, and the authors can address the above issues in a revision before a final editorial decision is made.

Authors’ response 20: Thank you for considering our manuscript a strong candidate for publication in *Nature Human Behaviour* and for your constructive comments aimed at enhancing our work. We have carefully addressed the issues you raised in our revision. We believe these changes have further improved the manuscript, and we hope these revisions meet your expectations.

References:

1. Landes, R. Zero-sum emotions and shame–honor dynamics. in *Honor and shame in Western history* (eds. Wettlaufer, J., Nash, D. & Hatlen, J. F.) 45–77 (Routledge, 2023).
2. Chowdhury, S. M. & Sheremeta, R. M. A generalized Tullock contest. *Public Choice* **147**, 413–420 (2011).
3. Heinz, M. & Schumacher, H. Signaling cooperation. *European Economic Review* **98**, 199–216 (2017).

4. Englmaier, F. & Gebhardt, G. Social dilemmas in the laboratory and in the field. *Journal of Economic Behavior & Organization* **128**, 85–96 (2016).
5. Rustagi, D., Engel, S. & Kosfeld, M. Conditional cooperation and costly monitoring explain success in forest commons management. *Science* **330**, 961–965 (2010).
6. Romano, A., Gross, J. & De Dreu, C. K. W. Conflict misperceptions between citizens and foreigners across the globe. *PNAS Nexus* **1**, pgac267 (2022).
7. Chiu, C.-Y., Gelfand, M. J., Yamagishi, T., Shteynberg, G. & Wan, C. Intersubjective culture: The role of intersubjective perceptions in cross-cultural research. *Perspect Psychol Sci* **5**, 482–493 (2010).
8. Vignoles, V. L. *et al.* Are Mediterranean societies “cultures of honor?”: Prevalence and implications of a cultural logic of honor across three world regions. *Pers Soc Psychol Bull* 01461672241295500 (2024) doi:10.1177/01461672241295500.

REVIEWER COMMENTS:

Reviewer 3

1. *The authors have done a thorough and conscientious job of addressing the issues I raised in the previous review. In particular, I now think the implications of the exploratory analyses are clearer than before, and I believe they convey useful information. I am happy to recommend publication of the paper. However, I have two additional editorial suggestions. I leave this to the editor, but I think if addressed, they would further enhance the overall readability of the article.*

First, although the implications of the exploratory analyses (pp. 14-15) are now better explained in the discussion, the purpose of these analyses was not clear when they were described for the first time. What is there is so descriptive that it isn't really clear why these analyses were carried out, and the reason is provided only in the discussion section. This makes it hard to understand the results. My suggestion is to add a sentence or so to each analysis, so that the purpose of the exploratory analysis is clear from the start. It would be a very minor adjustment, but would improve the readability of the paper quite a bit.

Authors' response 1: Thank you for this suggestion. We have revised the relevant sections in the Results to clarify the purpose of each exploratory analysis when it is first introduced:

“(Less-)efficient coordination success. To further shed light on the potential motives associated to the observed behavioural cooperation patterns, we compared the sum of individuals' own cooperation and expected partner's cooperation with two provision points of the public good. This allows to explore how the cultural logic of honour relates to individuals' anticipation of coordination success (see Methods).” (p. 15)

“(Less-)efficient competition. We also explored different forms of competition by subtracting expected partner's competition from individuals' own competition. This allows to distinguish different type of competitive behaviour which may have reflected different underlying motives (see Methods). Specifically, we explored how the cultural logic of honour relates to *efficient competition* (defined as spending just enough to win) and *less-efficient competition* (defined as overspending to make sure they win).” (pp. 16)

“(Un)conditional cooperation. By subtracting expected partner's cooperation from individuals' own cooperation, we also distinguished different types of cooperative behaviour (see Methods), and explored how the cultural logic of honour relates to *conditional cooperation* (defined as matching the expected contribution of one's partner in the same round) and *unconditional cooperation* (defined as exceeding the expected contribution of one's partner in the same round).” (p. 16)

2. *Second, I would add a table in the discussion section that summarizes the main findings. There are two very similar predictors (personal and perceived normative honour values) with two facets at two different levels of analysis. Competition and cooperation are predicted by these with additional cross-level interaction effects etc. etc. These are not easy to keep track of in mind, and having a table that succinctly displays which effects were significant and which was not would help readers interpret their implications. If space permits, this would*

enhance the impact of the paper in my view.

All in all, I think the article is a fine contribution to the literature. I am happy to support its publication.

Authors' response 2: We agree that including a summary table in the Discussion section improves the clarity and accessibility of our main findings. We have now added a new table (Table 2) that summarizes the associations between honour indicators at both the societal and individual levels and the outcome variables related to competition and cooperation:

Table 2. Support for hypotheses and summary of main findings

Predictor	Outcome	Competition		Cooperation	
		Hy.	Direction	Support	Direction
Societal-level honour	Behaviour	H1a	+*	Y	+*
	Expectation	H1b	+	N	+*
Individual-level honour					
Perceived normative honour values					
Self-promotion and retaliation	Behaviour	H2a	+*	Y	+**
Defence of family reputation			+**	Y	+
Self-promotion and retaliation	Expectation	H2b	+*	Y	+
Defence of family reputation			+**	Y	+***
Personal honour values					
Self-promotion and retaliation	Behaviour	H3	-	N	-
Defence of family reputation			+	N	+**
Cross-level interactions					
Personal honour (SPR) × Societal-level honour	Behaviour	/	+	/	+**
Personal honour (DFR) × Societal-level honour			-*	/	-*
Contextual effects	Behaviour	/	+	/	+
	Expectation	/	-	/	+

Note. Hy. = number of hypotheses, -/+ = direction of the effect, Y = hypothesis supported, N = hypothesis not supported (nonsignificant results). The contextual effects describe the differences in competition (or cooperation) among participants who have the same level of perceived normative and personal honour values but live in societies with different societal-level honour. * $p < .05$, ** $p < .01$, *** $p < .001$

Reviewer 4

1. Thank you very much for allowing me to read this fascinating manuscript. I found the manuscript easy to follow and applaud the authors' collective effort, including co-authors from different countries. I think this research is highly relevant, and the results are convincing. Furthermore, I appreciate all the included robustness checks, the transparency, and the illustration of results. The authors even included re-analyses of existing data sets, which can be found in the supplement. I replaced a former reviewer who was no longer available. Thus, I focused on whether the authors addressed the feedback sufficiently when reading the revised manuscript and the reviewers' comments. I added further remarks that the authors could address if they see fit.

The reviewers asked for a more detailed definition and description of central concepts. I

think the current version of this manuscript is thorough enough in that respect, but an example here and there might help understanding. For instance, since honor is a fundamental concept in their work, the authors could add one or two sentences, including examples of self- and other perceptions of honor. The same is true for honor-related norms or behaviors. Also, an example of a society more strongly characterized by a cultural logic of honor would positively influence readability.

Authors' response 1: Thank you for this suggestion. We have now adjusted the manuscript to more explicitly distinguish the individual and family facets of honour:

“Specifically, our measure of individual honour focused on valuing certain traits and actions (e.g., self-promotion, retaliation) to claim honour, whereas our measure of family honour mainly focused on protecting and defending the family’s reputation^{21,54}.” (p. 8)

We have also clarified that both personal honour values and perceived normative honour values refer to these same two facets:

“We operationalized the cultural logic of honour through the individual-level measures of personal endorsement of the abovementioned two facets of honour values (referred to as *personal values*) as well as intersubjective perceptions of how prevalent the two facets of honour values are within each society (referred to as *perceived normative values*)^{41,42}.” (p. 9)

Additionally, at the beginning of the Introduction section, we have provided examples of societies where honour has been found to be a core cultural value based on previous research, to help readers contextualize the concept of honour:

“*Honour*, a relevant yet underexplored cultural concept, is particularly prevalent in certain non-Western regions (e.g., the Middle Eastern and North African societies)¹⁸⁻²¹, and may act as an important cultural logic shaping how individuals navigate conflicts of interest between the self and others.” (p. 4)

We have also added information about the societies with the highest and lowest scores of societal-level honour in the Introduction section, based on data in the current study:

“The society mean of perceived normative honour values across both facets was used to construct a societal-level indicator, characterizing the extent to which a society can be considered a culture of honour (referred to as *societal-level honour*), ranging in our current samples from 4.44 (United States) to 6.03 (Egypt) (see Table 1 for scores of all samples).” (p. 9)

2. The manuscript was improved by adding more information on the ecological validity of the paradigms they used to address a reviewer's comment. I think it would help a reader unfamiliar with economic games if the authors would elaborate more on the kind of social interaction modelled in the two games they used (e.g., real-life examples).

Authors' response 2: Thank you for raising this point. We have added real-life examples to illustrate the types of social interactions modelled by the contest game and the step-level public goods game:

“Contest games are formally structured conflict situations in which one can only be better off at the cost of the other, and one risks being exploited if losing to one’s opponent^{43,44}. These games have been used to study informal and formal types of competition, as they model conflict situations that result in zero-sum outcomes (e.g., public debates, sports competitions, leadership elections).” (p. 6)

“Step-level public goods games (PGG) model situations where individuals can cooperate to achieve better collective outcomes at the risk of wasting personal efforts if coordination fails (e.g., building a neighbourhood security system or communal infrastructure)^{3,5}.” (pp. 6-7)

3. The authors indicate that their samples were stratified by age and gender. Could they say a few words when describing the samples and/or in the discussion section about the representativeness of their samples concerning other demographics, such as education, so that we get a better idea of the generalizability of the findings? Furthermore, if the samples are not representative of these demographics (the sample is likely disproportionate), can the authors briefly report previous findings on the relationship between these factors and honor (e.g., education)? The authors could also include a statement on whether the effects they found potentially under- or overestimate the true effect.

Authors’ response 3: In addition to age and gender, we also collected information on participants’ ethnic background, living environment, parental education level and subjective socioeconomic status (SES). We have now added a sentence in the Methods section to describe these characteristics so that readers can better assess the generalizability of our findings:

“Our sample was not stratified in terms of other demographic characteristics. The majority of participants self-identified as belonging to the majority ethnic group in the respective society (93.60%) and reported having an urban background (85.79%). Overall, participants reported a moderate level of parental education (i.e., above high school; $M = 4.33$, $SD = 1.58$) and subjective socioeconomic status ($M = 5.59$, $SD = 1.92$, on a scale from 1 to 10; see Table S35 for more information).” (pp. 23)

With regard to the potential relationship between these demographic characteristics and honour (and the observed effects), we decided not to speculate on whether our findings may over- or underestimate the true effects, in order to avoid any potential misinterpretation. There are two main reasons for this. First, our data showed that participants generally reported moderate levels of both objective (i.e., education) and subjective SES. Second, previous literature does not show a clear relationship between these sociodemographic factors and honour. In fact, the direction of the association between SES and honour values may depend on how SES is operationalized. For example, a recent study has shown that subjective SES is positively associated with honour endorsement, whereas objective SES is negatively associated (Sánchez-Rodríguez et al., 2025).

Sánchez-Rodríguez, Á., O’Dea, C., Uskul, A. K., Kirchner-Häusler, A., Vignoles, V., Chobthamkit, P., Achmad, R. A., Andrianto, S., Kristanto, A. A., Ardi, R., Lesmana, C. B. J., Castillo, V. A., Chaleeraktragoon, T., Zhi, A. C. H., Choompunuch, B., Cross, S. E., Nguyen, S. D., Fernandez, E. F., Purba, F. D., ... Uchida, Y. (2025). Overcoming low status or maintaining high status? A multinational examination of

the association between socioeconomic status and honour. *British Journal of Social Psychology*, 64, e12854. <https://doi.org/10.1111/bjso.12854>

4. *I was surprised to read that the authors excluded participants who did not identify as male or female. This was probably a pre-registered exclusion criterion, so I do not ask the authors to include them in their analyses now. As I noticed this issue independently of a former reviewer, I suggest the authors provide a rationale for each exclusion criterion in the supplementary materials (they already have one in the response letter). Furthermore, they could analyze potential gender differences (not only by adding gender as a control).*

Authors' response 4: Thank you for raising this important point. We have now provided a clear rationale for each exclusion criterion in the supplementary materials (see Table S2). As stated:

“We applied four exclusion criteria: (a) 120 participants who were not born in and currently located in the respective society; (b) 24 participants who did not self-identify as male or female. Participants were given a third gender option at the beginning of the study, and all were allowed to complete the survey and receive full compensation, including any game earnings. However, for data analysis, we only included participants who identified as male or female. This decision was made to align with our experimental design, which manipulated the gender of the game partner using binary categories (male, female, or gender not provided) to explore specific dynamics in competition and cooperation among female–female, male–male, and mixed-gender pairs. Given the scope of this manuscript, analyses of gender effects will be addressed in a separate paper; (c) 29 participants who failed the attention check question; and (d) 112 participants who failed all four comprehension questions designed to assess understanding of the contest game and step-level public goods game (PGG) rules.

This unregistered step was taken to improve the reliability of the data and the precision of the estimated associations between cultural variables and behavioural outcomes. Specifically, criterion (a) ensured that participants were embedded in the relevant cultural context; criterion (b) maintained alignment between the sample and the experimental manipulation; and criteria (c) and (d) ensured that participants were attentive and meaningfully engaged with the study.” (p. 3 in the SM)

5. *I appreciate the authors adding more information on their scales based on the reviewers' comments. However, some of these scales' purposes were still unclear to me. The authors could briefly explain why they included these scales when they mention them in the method section. For example, it makes sense that the authors included a relational mobility scale. Some countries neither cooperate nor compete, so people might be generally less willing to interact with strangers in these cultures.*

Authors' response 5: We had pre-registered the inclusion of the *belief in a zero-sum game* and *relational mobility* scales and originally provided our rationale in the supplementary materials. In response to your suggestion, we have now incorporated these explanations into the Methods section of the main manuscript to clarify the purpose of each scale:

“**Beliefs in a zero-sum game.** Beliefs in a zero-sum game captures the generalized beliefs about the nature of social relations involving completely conflicting interests⁵⁵.

Previous research has shown that this belief can lead to competition and conflict, and varies across societies and social economic status^{55,82}. To examine whether beliefs in a zero-sum game explain additional variation in competition beyond what could be explained by honour values, we measured this construct by asking participants to indicate the extent to which they agreed with eight statements reflecting the belief that life is a zero-sum game (e.g., “*The successes of some people are usually the failures of others*”; 1 = *strongly disagree* to 6 = *strongly agree*). Higher scores indicate stronger beliefs in a zero-sum game.

Relational mobility. Relational mobility is a socio-ecological variable that represents how much freedom and opportunity a society affords individuals to choose and dispose of interpersonal relationships based on personal preference⁵⁶. Past research has found higher levels of cooperation in societies characterized by more flexible and fluid social relations, as well as among individuals who perceive their environment as offering more opportunities to establish new relationships with strangers¹⁶. To examine whether relational mobility explain additional variation in cooperation beyond what could be explained by honour values, we measured this variable by asking participants to state how well 12 statements described the people in the society where they live (e.g., “*It is common for these people to have a conversation with someone they have never met before*”; 1 = *strongly disagree* to 6 = *strongly agree*). Higher scores indicate that people perceive their society to promote open and flexible social relations.” (pp. 29-30)

6. *The authors indicated that 112 participants failed all four comprehension questions in the main paper. Given the recent debate regarding the comprehensibility of economic paradigms (<https://www.sciencedirect.com/science/article/pii/S0167268125001581>), I think it would be interesting to learn how many participants failed one, two, and three comprehension checks and whether there have been differences between countries in that regard.*

Authors’ response 6: We have now included the percentage of participants who answered exactly one, two, or three out of four comprehension questions incorrectly (after two attempts) in Table S35 of the supplementary materials. We also tested whether these comprehension failure rates varied across societies. Our analyses revealed significant cross-societal differences, based on comparisons of model fit between models with and without society included as a random intercept. These findings have been added to the table note of Table S35:

“We found significant differences across societies in participants’ failure to answer the comprehension questions of economic games correctly. The between-society variance was significantly different from zero for the percentage of participants who failed exactly one question, $\chi^2(1) = 4.33, p = .037$; two questions, $\chi^2(1) = 6.34, p = .012$; and three questions, $\chi^2(1) = 19.10, p < .001$.” (p. 75 in the SM)

7. *Minor issues:*

This article includes a section called “results,” but several results were mentioned before that section.

Authors’ response 7: Thank you for pointing this out. We included a brief summary of the key findings at the end of the Introduction section to help guide readers and provide context for the subsequent sections. We found that this approach has also been used in other articles

published in *Nature Human Behaviour*. However, we are of course happy to remove or revise this paragraph if the editorial team recommends doing so.

8. *Typo p. 10 /ln. 214*

Authors' response 8: Thank you for spotting this typo. We have corrected the sentence by removing the extra words "as the."